# Refining the Martin–Hopkins method for estimating low-density lipoprotein cholesterol levels: Median versus optimal TG/VLDL-C ratio

**Jongseok Lee[1]\*, Hyelim Lee[1], Hwajung Cha[1], Jun Seok[2,3], In Cheol Jeong[1,4]\***

**1** School of Artificial Intelligence Convergence, Hallym University, Chuncheon, Republic of Korea, **2** College of Medicine, Hallym University, Chuncheon, Republic of Korea, **3** Division of Cardiology, Department of Internal Medicine, Myongji Hospital, Jecheon, Republic of Korea, **4** Cerebrovascular Disease Research Center, Hallym University, Chuncheon, Republic of Korea

\* ljs1844@hallym.ac.kr (JL); inchdol.jeong@hallym.ac.kr (IJ)

## Abstract

### Background

Low-density lipoprotein cholesterol (LDL-C), a major modifiable risk factor for cardiovascular diseases, is typically calculated using the Friedewald formula when triglyceride (TG) levels are below 400 mg/dL. Recent studies have demonstrated the superior accuracy of the Martin–Hopkins method across diverse populations. While this method estimates very-low-density lipoprotein cholesterol (VLDL-C) using strata-specific median TG/VLDL-C ratios, its reliance on median statistics raises questions about whether these ratios are truly optimal.

### Objectives and methods

This study evaluated the performance of the Martin–Hopkins method compared to the Friedewald formula, focusing on its potential for improvement by applying optimal TG/VLDL-C ratios. Using data from 18,322 individuals in the Korea National Health and Nutrition Examination Survey (KNHANES), we derived strata-specific optimal TG/VLDL-C ratios designed to maximize concordance with directly measured LDL-C values, based on LDL-C categories defined by clinical guidelines. We compared the performance of four LDL-C estimation models: the Friedewald formula ($LDL\text{-}C_F$), the original Martin–Hopkins method ($LDL\text{-}C_{M\text{-}N}$), and two alternative models that applied TG/VLDL-C ratios derived from our data—one using median values ($LDL\text{-}C_{KM\text{-}N}$) and the other using optimal values tailored to each stratum ($LDL\text{-}C_{KO\text{-}N}$).

### Results

The Martin–Hopkins method showed significantly higher concordance than the Friedewald formula for TG levels < 400 mg/dL (79.6% for $LDL\text{-}C_F$ vs. 83.2% for $LDL\text{-}C_{M\text{-}180}$, $p < 0.001$). Concordance improved by less than 2% for TG levels < 150 mg/dL (83.3%

**Data availability statement:** The data in this study are publicly accessible at [https://knhanes.cdc.go.kr/knhanes/index.do].

**Funding:** This work was supported by a National Research Foundation of Korea (NRF) grant funded by the Korean Government (MSIT) (No. 2022R1A5A8019303). The funders had no role in study design, data collection and analysis, decision to publish, or preparation of the manuscript.

**Competing interests:** The authors have declared that no competing interests exist.

vs. 84.9%), but by approximately 10% for TG levels of 150–399 mg/dL (68.8% vs. 78.0%). The largest discrepancy was observed in classifying LDL-C levels < 70 mg/dL among individuals with TG levels of 150–399 mg/dL (47.5% for LDL-C$_F$ vs. 90.3% for LDL-C$_{M-180}$). However, the overall concordance differed only modestly between the 10-cell and 180-cell Martin–Hopkins equations (82.8% for LDL-C$_{M-10}$ vs. 83.2% for LDL-C$_{M-180}$, a difference of 0.4%), indicating only a marginal benefit despite the substantial increase in the number of strata. Using optimal TG/VLDL-C ratios increased overall concordance compared to median ratios within the same stratification, with LDL-C$_{KO-N}$ estimates outperforming their LDL-C$_{KM-N}$ counterparts. However, this improvement was not statistically significant in LDL-C estimates derived from TG-only stratification.

## Conclusions

Applying optimal TG/VLDL-C ratios within the Martin–Hopkins method improves accuracy compared to median ratios, particularly when stratifications incorporate both TG and non–high-density lipoprotein cholesterol (non–HDL-C) levels. This enhancement can be achieved without increasing the number of strata, offering a practical pathway to refine LDL-C estimation while avoiding excessive stratification. Our findings suggest that while median statistics may be sufficient for TG-only stratifications, they do not fully capture optimal TG/VLDL-C ratios for combined TG and non–HDL-C stratifications.

## Introduction

Cardiovascular diseases (CVDs) are the world's leading cause of death, accounting for 32% of all global deaths in 2019 [1]. Low-density lipoprotein cholesterol (LDL-C)—a major modifiable risk factor for CVDs and the primary target in clinical practice guidelines—requires accurate assessment to guide therapeutic decisions and monitor treatment efficacy [2–4]. LDL-C is typically calculated using the Friedewald formula, a cost-effective alternative to beta quantification, the gold standard reference method. In the Korea National Health Screening Program (KNHSP), LDL-C is estimated using this formula when triglyceride (TG) levels are below 400 mg/dL. Among 14,024,331 participants screened in the 2015 KNHSP, 98% (13,743,272 individuals) had TG levels below this threshold [5], underscoring the formula's widespread application.

In their original study, Friedewald et al. [6] noted that LDL-C calculated using their formula was inaccurate for TG levels ≥ 400 mg/dL. However, accumulated evidence [7–10] suggests that even at TG levels < 400 mg/dL, the formula tends to underestimate LDL-C in individuals with elevated TG concentrations, particularly when such individuals are classified into lower LDL-C treatment categories [9–14]. For example, Lee et al. [12] reported that among individuals with TG levels of 200–399 mg/dL who had directly measured LDL-C levels ≥ 70 mg/dL, 55.4% were misclassified by the

Friedewald formula as having LDL-C < 70 mg/dL. This substantial rate of underestimation raises concerns that cardiovascular risk may be underestimated in individuals with moderate hypertriglyceridemia, potentially leading to missed opportunities for appropriate treatment in high-risk populations.

To address the limitations of the Friedewald formula, several alternative methods for LDL-C estimation have been developed [14–29]. In 2013, Martin et al. [14] introduced a new equation that improved LDL-C estimation accuracy, especially in classifying lower LDL-C levels in individuals with elevated triglycerides. Subsequent studies [30–35] have confirmed the superior accuracy of the Martin–Hopkins method across diverse populations and LDL-C measurement techniques, leading to its endorsement by various international guidelines and expert panels [36].

In contrast, Meeusen et al. [11] reported a marginal difference in overall concordance between the Friedewald and Martin–Hopkins equations (76.9% vs. 77.7%, a difference of only 0.8%) when compared to LDL-C measured by beta quantification. While the new method showed better concordance than the Friedewald formula in classifying LDL-C levels < 70 mg/dL and 70–99 mg/dL, it exhibited lower concordance in other classification ranges, raising questions about its practical utility in clinical settings.

The Friedewald formula estimates very-low-density lipoprotein cholesterol (VLDL-C) by assuming a fixed TG/VLDL-C ratio of 5 (in mg/dL). McNamara et al. [21] first noted that the TG/VLDL-C ratio varies depending on TG levels, a concept expanded by the Martin–Hopkins method, which incorporates both TG and non–high-density lipoprotein cholesterol (non–HDL-C) levels into a 180-cell stratification. This method applies strata-specific median TG/VLDL-C ratios for VLDL-C estimation. However, it remains unclear whether median values are the best representation of TG/VLDL-C ratios for accurate LDL-C estimation.

In this study, we derived strata-specific optimal TG/VLDL-C ratios—ranging from 1.0 to 10.0 in 0.1-unit increments—to maximize concordance between estimated and directly measured LDL-C levels based on guideline-defined LDL-C classification categories. To our knowledge, no prior studies have applied an optimal approach to TG/VLDL-C ratios within the Martin–Hopkins framework, nor systematically evaluated how closely median ratios approximate their optimal counterparts in terms of estimation accuracy.

We evaluated the performance of the Martin–Hopkins method compared to the Friedewald formula using data from 18,322 Koreans with LDL-C measured by a homogeneous enzymatic assay. A subsample of 11,930 individuals with TG distributions similar to those of the 2015 KNHSP cohort was selected to validate both equations, as well as other LDL-C estimation methods, such as the Sampson equation [15–29].

Using the stratification-based Martin–Hopkins method, we generated three LDL-C estimates: (1) LDL-C$_{M-N}$, using the original median TG/VLDL-C ratios from the N-cell table described by Martin et al. [14]; (2) LDL-C$_{KM-N}$, using median TG/VLDL-C ratios derived from our dataset; and (3) LDL-C$_{KO-N}$, using optimal TG/VLDL-C ratios derived from our dataset. These variations allowed for a direct comparison of estimation accuracy across different TG/VLDL-C ratios within the same stratification scheme, while also benchmarking against the Friedewald equation's fixed ratio of 5. Our primary objective was to assess the extent to which optimal ratios improve LDL-C estimation accuracy relative to median ratios.

Although fine-grained or exhaustive stratification may enhance estimation accuracy, it presents challenges for the reproducibility and external validation of the original 180-cell Martin–Hopkins equation. Prior studies [11,33], including our own, have generated 180-cell median TG/VLDL-C ratio matrices. However, they failed to replicate the consistent pattern reported by Martin et al. [14], in which the median ratios increased with rising TG levels and declining non–HDL-C levels within the same TG stratum. Excessive stratification may yield diminishing returns and increase the risk of overfitting to the derivation dataset. Notably, Martin et al. [14] observed in their study that the difference in overall concordance between the 10-cell and 180-cell schemes was only 1.2% (90.5% vs. 91.7%). Using simpler, lower-resolution stratification—while sacrificing a small degree of accuracy—may offer greater flexibility for incorporating novel yet potentially performance-enhancing factors. The optimal approach to TG/VLDL-C ratios within the Martin–Hopkins method could provide a practical pathway to improve LDL-C estimation accuracy without requiring excessive stratification.

## Materials and methods

### Study population

This study utilized data from the Korea National Health and Nutrition Examination Survey (KNHANES) spanning from 2009 to 2021. KNHANES is a nationwide, population-based, cross-sectional survey conducted annually by the Korea Disease Control and Prevention Agency (KDCA). The survey protocol was approved by the KDCA Institutional Review Board (approval codes: 2009-01CON-03-2C, 2010-02CON-21-C, 2011-02CON-06-C, 2012-01EXP-01-2C, 2013-07CON-03-4C, 2013-12EXP-03-5C, 2018-01-03-P-A, 2018-01-03-C-A, 2018-01-03-2C-A, and 2018-01-03-3C-A), and all participants provided written informed consent. The data used in this study were retrieved on January 15, 2023, and are publicly available at [https://knhanes.cdc.go.kr/knhanes/index.do]. As the dataset is fully de-identified, no information that could identify individual participants was accessible during or after data collection.

A total of 19,664 individuals with directly measured levels of total cholesterol (TC), high-density lipoprotein cholesterol (HDL-C), LDL-C, and triglycerides were initially considered. Participants with TG levels ≥ 400 mg/dL were excluded ($n = 1,342$; 6.8% of the total sample) to ensure the validity of the Friedewald formula, resulting in a final study sample of 18,322 individuals. For the years 2009–2011 and 2015, LDL-C levels were directly measured to represent the Korean population, while in 2012–2014 and 2016–2021, direct LDL-C measurements were restricted to participants with TG levels ≥ 200 mg/dL.

The study sample was divided into two groups: Population 1 (KNHANES 2009–2011 and 2015; $n = 11,930$) was used to validate LDL-C estimates and had a TG distribution similar to the national cohort of 14 million individuals screened in the 2015 KNHSP (S1 Table). Population 2 (KNHANES 2012–2014 and 2016–2021; $n = 6,392$) was combined with Population 1 to derive median and optimal TG/VLDL-C ratios, ensuring a sufficient sample of individuals with high TG levels.

### Lipid measurements and LDL-C estimation

Blood samples were collected from each participant's antecubital vein after an overnight fast of at least 8 hours. Serum lipid concentrations were directly measured using a homogeneous enzymatic assay with an automated analyzer (Hitachi Automatic Analyzer 7600, Hitachi, Tokyo, Japan). The measurements included TC (Pureauto S CHO-N; Sekisui Medical, Tokyo, Japan), HDL-C (Cholestest N HDL; Sekisui Medical), LDL-C (Cholestest N LDL; Sekisui Medical), and triglycerides (Pureauto S TG-N; Sekisui Medical). Non–HDL-C was calculated as TC minus HDL-C, and VLDL-C as non–HDL-C minus LDL-C.

Friedewald LDL-C (LDL-C$_F$) was calculated using the following formula [6]:

$$LDL-C_F \;=\; TC \;-\; HDL-C \;-\; (\text{triglycerides} \,/\, 5 \text{ in mg/dL})$$

Martin–Hopkins LDL-C (LDL-C$_{M-N}$) was calculated using the following equation [14]:

$$LDL-C_{M-N} \;=\; (\text{non–HDL-C}) \;-\; (\text{triglycerides} \,/\, AF_N \text{ in mg/dL})$$

where $AF_N$ is an adjustable factor determined as the median TG/VLDL-C ratio based on N-cell stratification. LDL-C$_{M-10}$ and LDL-C$_{M-180}$ were calculated using the strata-specific median TG/VLDL-C ratios in the 10-cell and 180-cell tables described by Martin et al. [14], respectively. For consistency with the original method, the numerical subscripts for LDL-C$_{M-10}$ and LDL-C$_{M-180}$ were retained, even though individuals with TG levels ≥ 400 mg/dL were excluded from the analysis.

The main focus of this study was to compare the performance of optimal TG/VLDL-C ratios with median TG/VLDL-C ratios in LDL-C estimation. For this purpose, two variations of LDL-C estimates were calculated using strata-specific TG/VLDL-C ratios derived from our dataset:

- LDL-C$_{KM-N}$: Based on median TG/VLDL-C ratios, where "$_{KM}$" indicates median values derived from our Korean sample, and "$_N$" specifies the number of strata.

- LDL-C$_{KO-N}$: Based on optimal TG/VLDL-C ratios, with "$_O$" in "$_{KO-N}$" denoting optimal values.

The optimal TG/VLDL-C ratio was designed to maximize concordance between estimated and directly measured LDL-C levels, based on the National Cholesterol Education Program Adult Treatment Panel III (NCEP–ATP III) guideline classification. When stratification was based on TG levels alone, rather than combined TG and non–HDL-C levels, the subscript "$_{TG}$" was added, as in "LDL-C$_{KM-N-TG}$" or "LDL-C$_{KO-N-TG}$." Alternative LDL-C estimates were also calculated using previously published equations [15–29], in addition to the Friedewald formula and the Martin–Hopkins method.

To assess the contribution of each stratification parameter to LDL-C estimation, we first performed stratification using triglycerides alone, followed by the inclusion of non–HDL-C levels. Strata-specific tables of median and optimal TG/VLDL-C ratios were generated based on our dataset, using 6, 10, 12, 28, and 180 cells. However, due to an insufficient sample size, optimal TG/VLDL-C ratios could not be derived for the 180-cell table. TG levels were initially divided into six categories (< 50, 50–99, 100–149, 150–199, 200–299, and 300–399 mg/dL). These categories were then expanded in two ways for LDL-C estimation: one approach introduced non–HDL-C cutoff points into the six TG categories, while the other further stratified TG levels without considering non–HDL-C levels.

## Data analysis

Categorical variables were summarized as frequencies and percentages, while numerical variables were presented as medians and interquartile ranges (IQRs). Comparisons between Population 1 and Population 2 were conducted using Fisher's exact test for categorical variables, and the median test for numerical variables. Linear regression analysis was performed to assess the extent to which VLDL-C could be explained by triglycerides, non–HDL-C, HDL-C, sex, and age.

LDL-C values, including both directly measured (LDL-C$_D$) and estimated LDL-C (LDL-C$_E$), were classified according to the NCEP–ATP III guideline into six categories (< 70, 70–99, 100–129, 130–159, 160–189, and ≥ 190 mg/dL). The concordance rate was defined as the proportion of cases in which LDL-C$_D$ fell into the same category as the corresponding LDL-C$_E$, with initial classification based on the estimated values. Cross-tabulations were constructed to assess concordance between each LDL-C$_E$ and LDL-C$_D$. McNemar's exact test for correlated proportions was used to compare concordance rates between different LDL-C estimation methods. The 95% confidence intervals for concordance rates were calculated using the Clopper–Pearson exact method.

Bland–Altman analysis was performed to assess the agreement between LDL-C$_E$ and LDL-C$_D$. For three estimation methods (LDL-C$_F$, LDL-C$_{M-180}$, and LDL-C$_{KO-28}$), the bias (difference between LDL-C$_E$ and LDL-C$_D$) was plotted against the mean of the two values. The mean bias and the 95% limits of agreement (mean ± 1.96 SD) were calculated to evaluate systematic differences and dispersion across the range of LDL-C values. In addition, model fit and error metrics were assessed for each LDL-C estimation method in comparison with LDL-C$_D$, including the coefficient of determination ($R^2$), mean relative error (MRE), mean absolute error (MAE), and mean squared error (MSE).

Reclassification was defined as a shift in the guideline-based treatment category when using an alternative LDL-C estimation method or directly measured LDL-C, either upward (to a higher category) or downward (to a lower category), compared with LDL-C$_F$. The correctness of each reclassification was evaluated against directly measured LDL-C. To evaluate the clinical implications of LDL-C estimation methods, a reclassification analysis was conducted to quantify the improvement of the Martin–Hopkins method over the Friedewald formula in correctly assigning individuals to treatment categories based on LDL-C$_D$.

Strata-specific optimal TG/VLDL-C ratios were derived from our dataset by assessing ratios ranging from 1.0 to 10.0 in 0.1-unit increments, using Python (version 3.11.9) for computations. All statistical analyses were performed using SPSS for Windows (version 26.0; SPSS Inc., Chicago, IL, USA), with a two-tailed $p$-value < 0.05 considered statistically significant.

## Results

### Study samples

Table 1 summarizes the sex, age, and lipid characteristics of the study population. Population 1 ($n = 11,930$), which was used to validate LDL-C estimates, included 1,500 individuals (12.6%) with TG levels of 200–399 mg/dL. In contrast, Population 2 ($n = 6,392$), which was combined with Population 1 to derive median and optimal TG/VLDL-C ratios, consisted almost entirely of individuals within the same TG range, with 6,384 individuals (99.9%), as shown in S1 Table. Significant differences between the two populations were observed for all variables, except for the median LDL-C$_F$. The median (IQR) TG/VLDL-C ratio was 5.1 (4.0–6.3) in Population 1 and 6.3 (5.3–7.4) in Population 2 ($p < 0.001$).

### Relationship of VLDL-C with non–HDL-C, HDL-C, sex, and age

Table 2 presents the results of the regression analysis examining factors associated with VLDL-C levels among individuals with TG levels < 400 mg/dL. In Model 1, triglycerides alone explained 64.5% of the variance in VLDL-C. When non–HDL-C was added as an additional predictor in Model 2, the explained variance increased slightly to 66.7%. In this model, the

**Table 1. Characteristics of the study population: 18,322 Koreans aged 10 to 80 years with triglyceride levels < 400 mg/dL, based on data from the KNHANES 2009–2021.**

| | Number (Percentage) or Median (IQR) a | | | | | | |
| --- | --- | --- | --- | --- | --- | --- | --- |
| | Total | | Population 1 b | | Population 2 b | | |
| Variable | (n = 18,322) | | (n = 11,930) | | (n = 6,392) | | p-value c |
| Sex, n (%) | | | | | | | |
| Male | 9,443 | (51.5) | 5,571 | (46.7) | 3,872 | (60.6) | < 0.001 |
| Female | 8,879 | (48.5) | 6,359 | (53.3) | 2,520 | (39.4) | |
| Age, years | 48 | (34–61) | 45 | (29–59) | 53 | (41–63) | < 0.001 |
| 10–18 | 1,483 | (8.1) | 1,304 | (10.9) | 179 | (2.8) | < 0.001 |
| 19–80 | 16,839 | (91.9) | 10,626 | (89.1) | 6,213 | (97.2) | |
| Cholesterol, mg/dL | | | | | | | |
| Total | 189 | (165–216) | 182 | (159–207) | 204 | (179–231) | < 0.001 |
| HDL-C | 46 | (39–54) | 49 | (42–57) | 42 | (37–48) | < 0.001 |
| LDL-C$_D$ | 112 | (91–135) | 109 | (89–131) | 118 | (96–142) | < 0.001 |
| LDL-C$_F$ | 108 | (87–132) | 108 | (90–131) | 108 | (85–134) | 0.495 |
| LDL-C$_{M-180}$ | 114 | (93–137) | 110 | (90–132) | 121 | (100–145) | < 0.001 |
| Non–HDL-C | 142 | (116–169) | 132 | (109–158) | 161 | (138–187) | < 0.001 |
| VLDL-C | 27 | (18–40) | 21 | (15–29) | 41 | (33–50) | < 0.001 |
| Triglycerides, mg/dL | 156 | (86–237) | 102 | (69–150) | 247 | (219–291) | < 0.001 |
| TG/ VLDL-C ratio | 5.6 | (4.4–6.8) | 5.1 | (4.0–6.3) | 6.3 | (5.3–7.4) | < 0.001 |

KNHANES: Korea National Health and Nutrition Examination Survey; IQR: interquartile range; HDL-C: high-density lipoprotein cholesterol; LDL-C: low-density lipoprotein cholesterol; LDL-C$_D$: directly measured LDL-C using the homogeneous enzymatic assay; LDL-C$_F$: LDL-C calculated using the Friedewald formula; LDL-C$_{M-180}$: LDL-C calculated using the original 180-cell Martin–Hopkins equation proposed by Martin et al. [14]; non–HDL-C: non–high-density lipoprotein cholesterol; VLDL-C: very-low-density lipoprotein cholesterol; TG/VLDL-C ratio: ratio of triglycerides to very-low-density lipoprotein cholesterol.

[a]Categorical variables are expressed as frequencies (percentages), and continuous variables are expressed as medians (IQRs).

[b]Population 1 was used to validate the LDL-C estimates, while Population 2 was combined with Population 1 to derive median and optimal TG/VLDL-C ratios.

[c]Statistical significance was assessed using Fisher's exact test for categorical variables and the median test for continuous variables to compare Population 1 and Population 2.

**Table 2. Regression analysis results for VLDL-C as the dependent variable in 11,930 Koreans (Population 1) aged 10 to 80 years with triglyceride levels < 400 mg/dL.**

| Model | Independent variable | Dependent variable: VLDL-C | | | | | |
|---|---|---|---|---|---|---|---|
| | | *b* | (95% CI) | *p*-value | *R* | Partial *R* | Adjusted *R²* |
| Model 1 | | | | | | | 0.645 |
| | Triglycerides (mg/dL) | 0.136 | (0.134, 0.138) | < 0.001 | 0.803 | 0.803 | |
| | Intercept | 6.910 | (6.659, 7.161) | < 0.001 | | | |
| Model 2 | | | | | | | 0.667 |
| | Triglycerides (mg/dL) | 0.123 | (0.121, 0.125) | < 0.001 | 0.803 | 0.748 | |
| | Non–HDL-C (mg/dL) | 0.055 | (0.051, 0.059) | < 0.001 | 0.495 | 0.247 | |
| | Intercept | 1.061 | (0.583, 1.540) | < 0.001 | | | |
| Model 3 | | | | | | | 0.695 |
| | Triglycerides (mg/dL) | 0.127 | (0.124, 0.129) | < 0.001 | 0.803 | 0.741 | |
| | Non–HDL-C (mg/dL) | 0.041 | (0.037, 0.045) | < 0.001 | 0.495 | 0.192 | |
| | HDL-C (mg/dL) | 0.078 | (0.068, 0.089) | < 0.001 | −0.248 | 0.130 | |
| | Sex [a] | 0.864 | (0.621, 1.108) | < 0.001 | −0.075 | 0.035 | |
| | Age (years) | 0.097 | (0.090, 0.104) | < 0.001 | 0.366 | 0.252 | |
| | Intercept | −6.209 | (−6.992, −5.425) | < 0.001 | | | |

VLDL-C: very-low-density lipoprotein cholesterol; Non–HDL-C: non–high-density lipoprotein cholesterol; HDL-C: high-density lipoprotein cholesterol; *b*: unstandardized beta coefficient; CI: confidence interval; *R*: Pearson correlation coefficient; Partial *R*: partial correlation coefficient.

[a]Sex was coded as 0 for male and 1 for female.

partial $R^2$ for triglycerides was $(0.748)^2 = 0.560$, while that for non–HDL-C was $(0.247)^2 = 0.061$, indicating that triglycerides accounted for approximately 9.2 times more of the variance in VLDL-C than did non–HDL-C.

Model 3 was further adjusted for HDL-C, sex, and age in addition to non–HDL-C. Triglycerides remained the strongest predictor of VLDL-C, with a partial $R = 0.741$. Among the additional covariates, the relative explanatory power for VLDL-C was ranked by partial $R$ values as follows: age (0.252), non–HDL-C (0.192), HDL-C (0.130), and sex (0.035).

## Median versus Optimal TG/VLDL-C ratios

In this study, strata-specific optimal TG/VLDL-C ratios were determined by testing values ranging from 1.0 to 10.0 in 0.1-unit increments, aiming to maximize concordance between estimated and directly measured LDL-C levels according to the NCEP–ATP III guideline classification. As shown in Fig 1, concordance and discordance rates were plotted across the full range of TG/VLDL-C ratios for each of the six strata. An optimal TG/VLDL-C ratio was identified for each TG stratum, with values increasing progressively with TG levels: (A) 3.2 for TG levels < 50 mg/dL; (B) 4.6 for 50–99 mg/dL; (C) 5.6 for 100–149 mg/dL; (D) 5.9 for 150–199 mg/dL; (E) 6.3 for 200–299 mg/dL; and (F) 6.4 for 300–399 mg/dL. As the TG/VLDL-C ratio increased from 1.0 to 10.0, the under-classification rate consistently decreased, whereas the over-classification rate increased—demonstrating a trade-off between these two types of discordance.

As shown in Fig 1, the optimal TG/VLDL-C ratios consistently outperformed the fixed ratio of 5 used in the Friedewald formula, particularly in strata with higher TG levels. For TG levels of 200–299 mg/dL (Fig 1E), the concordance rate improved from 63.5% (fixed ratio of 5) to 75.3% (optimal ratio of 6.3), yielding an 11.8-percentage-point increase. For TG levels of 300–399 mg/dL (Fig 1F), the concordance rose markedly from 49.0% to 72.0% using the optimal ratio of 6.4, yielding an absolute improvement of 23.0 percentage points.

Table 3 presents the median and optimal TG/VLDL-C ratios based on two different stratification schemes: (1) a 6-cell scheme based solely on TG levels, and (2) a 12-cell scheme incorporating a non–HDL-C cutoff of 130 mg/dL within each TG stratum. In both stratification schemes, median and optimal TG/VLDL-C ratios increased progressively with rising TG

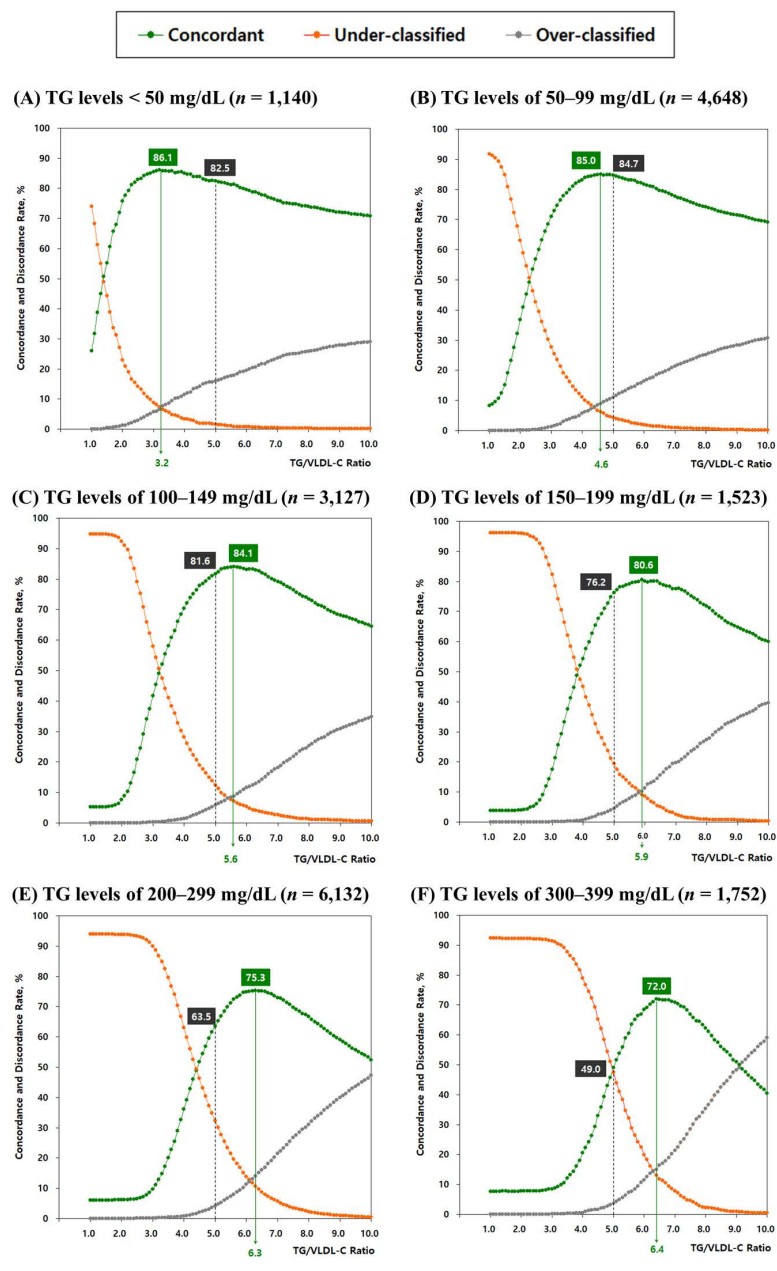

**Fig 1. Optimal TG/VLDL-C ratios across six TG strata.** Panels (A–F) depict the concordance (green line), under-classification (orange line), and over-classification (gray line) rates based on the NCEP–ATP III guideline classification as a function of TG/VLDL-C ratio across six TG strata. In each panel, the vertical green line represents the TG/VLDL-C ratio that achieved the highest concordance (i.e., the optimal value). Dashed black lines indicate the fixed ratio of 5 used in the Friedewald formula. **Abbreviations:** TG/VLDL-C ratio: ratio of triglycerides to very-low-density lipoprotein cholesterol; TG: triglyceride; NCEP–ATP III: National Cholesterol Education Program Adult Treatment Panel III.

levels. However, within the same TG stratum of the 12-cell scheme, both ratios tended to decrease when non–HDL-C levels were elevated.

In the 6-cell stratification (first column of Table 3), the median and optimal TG/VLDL-C ratios were nearly identical. In contrast, the 12-cell stratification (middle and last columns of Table 3) showed more pronounced discrepancies between

**Table 3. Median (95% CI) and optimal TG/VLDL-C ratios: TG strata (6-cell) and combined TG and non–HDL-C strata (12-cell).**

| TGs, mg/dL | TG/VLDL-C ratio | | | | | | | | |
|---|---|---|---|---|---|---|---|---|---|
| | 6-cell strata | | | 6 of 12-cell strata: non–HDL-C < 130 mg/dL | | | 6 of 12-cell strata: non–HDL-C ≥ 130 mg/dL | | |
| | n | Median (95% CI, ACL) [a] | Optimal [b] | n | Median (95% CI, ACL) [a] | Optimal [b] | n | Median (95% CI, ACL) [a] | Optimal [b] |
| < 50 | 1,140 | **3.1** (2.98–3.19, 95.3%) | **3.2** | 949 | **3.2** (3.07–3.29, 95.6%) | **3.9** | 191 | **2.7** (2.58–2.89, 95.8%) | **2.9** |
| 50–99 | 4,648 | **4.4** (4.39–4.49, 95.2%) | **4.6** | 2,945 | **4.6** (4.48–4.64, 95.3%) | **4.8** | 1,703 | **4.2** (4.15–4.32, 95.3%) | **4.5** |
| 100–149 | 3,127 | **5.4** (5.36–5.49, 95.1%) | **5.6** | 1,211 | **5.8** (5.63–5.84, 95.6%) | **6.1** | 1,916 | **5.2** (5.10–5.26, 95.3%) | **5.2** |
| 150–199 | 1,523 | **5.9** (5.76–5.95, 95.4%) | **5.9** | 400 | **6.4** (6.26–6.60, 95.0%) | **6.4** | 1,123 | **5.7** (5.58–5.78, 95.1%) | **5.9** |
| 200–299 | 6,132 | **6.2** (6.15–6.25, 95.2%) | **6.3** | 1,217 | **6.9** (6.85–7.05, 95.5%) | **6.8** | 4,915 | **6.0** (5.99–6.07, 95.1%) | **6.1** |
| 300–399 | 1,752 | **6.4** (6.36–6.52, 95.3%) | **6.4** | 221 | **7.3** (7.09–7.44, 95.7%) | **8.0** | 1,531 | **6.3** (6.22–6.38, 95.4%) | **6.4** |

TG/VLDL-C ratio: ratio of triglycerides to very-low-density lipoprotein cholesterol; TG: triglyceride; non–HDL-C: non–high-density lipoprotein cholesterol; CI: confidence interval; ACL: actual confidence level.

[a]The 95% confidence interval for the median was constructed without assuming any specific distribution of the TG/VLDL-C ratio. The actual coverage may exceed 95%.

[b]The optimal TG/VLDL-C ratio was defined as the value that maximized concordance between estimated and directly measured LDL-C levels, based on the classification criteria of the National Cholesterol Education Program Adult Treatment Panel III (NCEP–ATP III) guideline, using either TG strata (6-cell) or combined TG and non–HDL-C strata (12-cell).

the median and optimal values, particularly in subgroups with lower TG levels and non–HDL-C levels < 130 mg/dL. For example, among individuals with TG levels < 50 mg/dL and non–HDL-C < 130 mg/dL, the optimal TG/VLDL-C ratio (3.9) was notably higher than the median value (3.2). Additionally, the 95% confidence intervals for the median ratios were non-overlapping, indicating statistically significant differences across the groups.

To further evaluate the effect of TG stratification alone on LDL-C estimation accuracy, we additionally derived median and optimal TG/VLDL-C ratios using a 12-cell scheme based solely on TG levels (S2 Table). While the median ratios consistently increased with higher TG levels, this trend did not hold consistently for the optimal ratios. For comparison purposes, we additionally calculated median TG/VLDL-C ratios from our dataset using the 10-cell and 180-cell stratification schemes reported by Martin et al. [14] (S3 and S4 Tables, respectively).

Fig 2 presents a 28-cell stratification scheme based on both TG and non–HDL-C levels, with each cell containing at least 100 samples (see S5 Table for details). This visualization demonstrates the extent to which TG/VLDL-C ratios vary according to TG and non–HDL-C strata and highlights notable discrepancies between the median and optimal values across the 28-cell grid.

## Comparison of overall concordance

Table 4 summarizes the overall concordance between estimated and directly measured LDL-C among individuals with TG levels < 400 mg/dL. The statistical significance of pairwise differences in concordance between LDL-C estimates was assessed using McNemar's exact test (see S6 Table for details).

The overall concordance was 79.6% for LDL-C$_F$, 82.8% for LDL-C$_{M-10}$, and 83.2% for LDL-C$_{M-180}$ ($p < 0.001$ for each comparison with LDL-C$_F$). While LDL-C$_{M-180}$ showed significantly higher overall concordance than LDL-C$_{M-10}$, the difference was modest, at only 0.4% ($p = 0.045$). Both LDL-C$_{M-10}$ and LDL-C$_{M-180}$ were calculated using the original median TG/VLDL-C ratios from the 10-cell and 180-cell tables reported by Martin et al. [14].

When the median TG/VLDL-C ratios derived from our dataset were applied to the same stratification schemes, overall concordance rates were 82.8% for LDL-C$_{KM-10}$ and 83.7% for LDL-C$_{KM-180}$, with a 0.9% improvement ($p < 0.001$). Although LDL-C$_{M-10}$ and LDL-C$_{KM-10}$ yielded identical concordance rates (both 82.8%), LDL-C$_{KM-180}$ showed significantly higher concordance than LDL-C$_{M-180}$ (83.7% vs. 83.2%, $p = 0.003$).

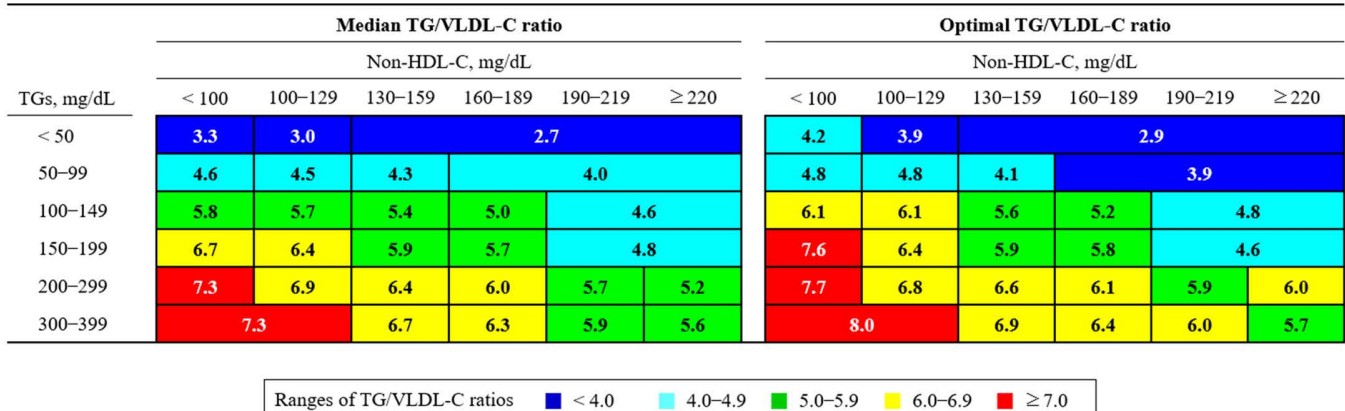

**Fig 2. Median and optimal TG/VLDL-C ratios by TG and non–HDL-C strata (28-cell).** The left and right panels display the median and optimal TG/VLDL-C ratios, respectively. Color bands represent the TG/VLDL-C ratio ranges as follows: blue (< 4.0), light blue (4.0–4.9), green (5.0–5.9), yellow (6.0–6.9), and red (≥ 7.0). **Abbreviations:** TG/VLDL-C ratio: ratio of triglycerides to very-low-density lipoprotein cholesterol; TG: triglyceride; non–HDL-C: non–high-density lipoprotein cholesterol.

**Table 4. Concordance of Friedewald and Martin–Hopkins LDL-C estimates with directly measured LDL-C according to the NCEP–ATP III guideline, overall and stratified by triglyceride levels.**

| | Triglyceride category, mg/dL | | | | | | | |
|---|---|---|---|---|---|---|---|---|
| | < 50 | 50–99 | 100–149 | 150–199 | 200–299 | 300–399 | Overall | |
| LDL-C$_E$ | CN (%) | CN (%) | CN (%) | CN (%) | CN (%) | CN (%) | CN (%) | p-value [a] |
| LDL-C$_F$ | 940 (82.5) | 3,934 (84.7) | 2,550 (81.6) | 1,159 (76.4) | 734 (65.4) | 183 (48.4) | 9,500 (79.6) | N/A |
| LDL-C$_{M-10}$ | 958 (84.0) | 3,941 (84.8) | 2,623 (83.9) | 1,222 (80.5) | 854 (76.1) | 276 (73.0) | 9,874 (82.8) | 0.045 |
| LDL-C$_{M-180}$ | 978 (85.8) | 3,957 (85.2) | 2,635 (84.3) | 1,225 (80.7) | 866 (77.2) | 264 (69.8) | 9,925 (83.2) | |
| LDL-C$_{KM-6-TG}$ [b] | 979 (85.9) | 3,934 (84.7) | 2,617 (83.7) | 1,224 (80.6) | 870 (77.5) | 274 (72.5) | 9,898 (83.0) | 0.089 |
| LDL-C$_{KO-6-TG}$ [b] | 981 (86.1) | 3,948 (85.0) | 2,627 (84.1) | 1,224 (80.6) | 870 (77.5) | 274 (72.5) | 9,924 (83.2) | |
| LDL-C$_{KM-10}$ | 959 (84.1) | 3,924 (84.4) | 2,630 (84.2) | 1,227 (80.8) | 865 (77.1) | 270 (71.4) | 9,875 (82.8) | 0.013 |
| LDL-C$_{KO-10}$ | 948 (83.2) | 3,963 (85.3) | 2,638 (84.4) | 1,235 (81.4) | 866 (77.2) | 277 (73.3) | 9,927 (83.2) | |
| LDL-C$_{KM-12-TG}$ [b] | 985 (86.4) | 3,948 (85.0) | 2,617 (83.7) | 1,224 (80.6) | 870 (77.5) | 274 (72.5) | 9,918 (83.1) | 0.087 |
| LDL-C$_{KO-12-TG}$ [b] | 992 (87.0) | 3,954 (85.1) | 2,632 (84.2) | 1,231 (81.1) | 868 (77.4) | 277 (73.3) | 9,954 (83.4) | |
| LDL-C$_{KM-12}$ | 983 (86.2) | 3,950 (85.0) | 2,627 (84.1) | 1,227 (80.8) | 856 (77.2) | 274 (72.5) | 9,927 (83.2) | 0.006 |
| LDL-C$_{KO-12}$ | 990 (86.8) | 3,963 (85.3) | 2,638 (84.4) | 1,235 (81.4) | 873 (77.8) | 278 (73.5) | 9,977 (83.6) | |
| LDL-C$_{KM-28}$ | 972 (85.3) | 3,944 (84.9) | 2,635 (84.3) | 1,242 (81.8) | 873 (77.8) | 284 (75.1) | 9,950 (83.4) | < 0.001 |
| LDL-C$_{KO-28}$ | 992 (87.0) | 3,967 (85.4) | 2,652 (84.9) | 1,247 (82.1) | 876 (78.1) | 286 (75.7) | 10,020 (84.0) | |
| LDL-C$_{KM-180}$ | 972 (85.3) | 3,968 (85.4) | 2,649 (84.8) | 1,242 (81.8) | 873 (77.8) | 282 (74.6) | 9,986 (83.7) | N/A |

NCEP–ATP III: National Cholesterol Education Program Adult Treatment Panel III; LDL-C: low-density lipoprotein cholesterol; LDL-C$_E$: estimated LDL-C; CN: concordant number; LDL-C$_F$: LDL-C calculated using the Friedewald formula; LDL-C$_{M-N}$ (LDL-C$_{M-10}$ and LDL-C$_{M-180}$): LDL-C calculated using the N-cell tables with the original median ratios of triglycerides to very-low-density lipoprotein cholesterol (TG/VLDL-C) reported by Martin et al. [14]; LDL-C$_{KM-N}$ (LDL-C$_{KM-6-TG}$, LDL-C$_{KM-10}$, LDL-C$_{KM-12-TG}$, LDL-C$_{KM-12}$, LDL-C$_{KM-28}$, and LDL-C$_{KM-180}$): LDL-C calculated using the N-cell tables with the median TG/VLDL-C ratios derived from our dataset; LDL-C$_{KO-N}$ (LDL-C$_{KO-6-TG}$, LDL-C$_{KO-10}$, LDL-C$_{KO-12-TG}$, LDL-C$_{KO-12}$, and LDL-C$_{KO-28}$): LDL-C calculated using the N-cell tables with the optimal TG/VLDL-C ratios derived from our dataset; N/A: not applicable.

[a]Statistical significance of differences in overall concordance between two LDL-C estimates was assessed using McNemar's exact test for correlated proportions.

[b]When stratification was based on TG levels alone—rather than combined TG and non–HDL-C levels—the subscript "$_{TG}$" was added, as in LDL-C$_{KM-N-TG}$ or LDL-C$_{KO-N-TG}$.

As shown in Table 4, the use of optimal TG/VLDL-C ratios led to higher overall concordance than the use of median TG/VLDL-C ratios within the same stratification schemes. However, in the stratification schemes based solely on triglycerides, the differences were not statistically significant: 83.0% for LDL-C$_{KM-6-TG}$ vs. 83.2% for LDL-C$_{KO-6-TG}$ ($p = 0.089$) and 83.1% for LDL-C$_{KM-12-TG}$ vs. 83.4% for LDL-C$_{KO-12-TG}$ ($p = 0.087$). By contrast, in all stratification schemes incorporating both TG and non–HDL-C levels, LDL-C$_{KO-N}$ estimates showed significantly higher overall concordance than their corresponding LDL-C$_{KM-N}$ estimates. These included: 82.8% for LDL-C$_{KM-10}$ vs. 83.2% for LDL-C$_{KO-10}$ ($p = 0.013$); 83.2% for LDL-C$_{KM-12}$ vs. 83.6% for LDL-C$_{KO-12}$ ($p = 0.006$); and 83.4% for LDL-C$_{KM-28}$ vs. 84.0% for LDL-C$_{KO-28}$ ($p < 0.001$).

Alternative LDL-C estimates were also calculated using previously published equations. S7 Table summarizes the overall concordance rates of these equations for LDL-C estimation in individuals with TG levels < 400 mg/dL. Among the 17 LDL-C equations assessed, the 180-cell Martin–Hopkins equation demonstrated the highest overall concordance (83.2%), followed by the Sampson (82.2%), Rao (81.9%), Puavilai (81.4%), Chen (81.4%), DeLong (80.8%), and Friedewald (79.6%) equations. All other equations demonstrated lower overall concordance than the Friedewald formula.

In addition to overall concordance rates, model fit and error metrics were evaluated for each LDL-C estimation method in comparison with directly measured LDL-C, as summarized in S8 Table. Methods are ranked in descending order of overall concordance. LDL-C$_{KO-28}$ yielded the lowest mean absolute error (5.09 mg/dL), smallest mean squared error (47.64 mg/dL$^2$), and lowest mean relative error (0.73%) while achieving the highest $R^2$ value (0.954). The 180-cell Martin–Hopkins equation also demonstrated excellent performance, with a mean absolute error of 5.11 mg/dL and $R^2$ of 0.952. These results support the superiority of our optimized method (LDL-C$_{KO-28}$) over both the Friedewald formula and other alternatives.

## Concordance by TG and estimated LDL-C levels

Fig 3 illustrates the concordance rates of LDL-C estimates calculated using the Friedewald and Martin–Hopkins methods, compared with directly measured LDL-C, based on the NCEP–ATP III guideline classification. The results are stratified by TG levels, with Fig 3A showing concordance for TG levels < 150 mg/dL and Fig 3B for TG levels of 150–399 mg/dL. LDL-C estimates based on the Martin–Hopkins method included not only LDL-C$_{M-180}$, calculated using the original 180-cell Martin–Hopkins equation, but also LDL-C$_{KO-6}$, LDL-C$_{KO-12}$, and LDL-C$_{KO-28}$, calculated using the optimal TG/VLDL-C ratios derived from our dataset. The statistical significance of differences in concordance between these LDL-C estimates is detailed in S9 and S10 Tables.

Among individuals with TG levels < 150 mg/dL (Fig 3A), all four Martin–Hopkins-based estimates showed statistically significantly higher concordance than the Friedewald-calculated LDL-C (LDL-C$_F$); however, the magnitude of improvement was modest, with increases of approximately 2%. In contrast, among individuals with TG levels of 150–399 mg/dL (Fig 3B), each of the Martin–Hopkins-based estimates demonstrated a substantial improvement over LDL-C$_F$, with concordance higher by approximately 10% ($p < 0.001$ for all comparisons).

Concordance rates by six TG levels are presented in Table 4. LDL-C$_F$ showed a declining trend in concordance with increasing TG levels. For individuals with TG levels of 200–399 mg/dL, the concordance was only 61.1% for LDL-C$_F$, broken down as follows: 65.4% for TG levels of 200–299 mg/dL and 48.4% for TG levels of 300–399 mg/dL. In these TG categories, LDL-C estimates calculated by the Martin–Hopkins method—including LDL-C$_{M-N}$, LDL-C$_{KM-N}$, and LDL-C$_{KO-N}$—demonstrated marked improvements in concordance compared to LDL-C$_F$. These LDL-C estimates achieved concordance rates exceeding 75% for TG levels of 200–299 mg/dL and over 70% for TG levels of 300–399 mg/dL, with the exception of LDL-C$_{M-180}$.

To further assess the agreement between estimated LDL-C (LDL-C$_E$) and directly measured LDL-C (LDL-C$_D$), Bland–Altman analyses were conducted for three estimation methods: LDL-C$_F$, LDL-C$_{M-180}$, and LDL-C$_{KO-28}$. As shown in Fig 4, each panel displays the bias (i.e., the difference between LDL-C$_E$ and LDL-C$_D$) plotted against the mean of the two values. The black dashed line indicates the mean bias, while the orange and red dashed lines represent the upper and lower 95%

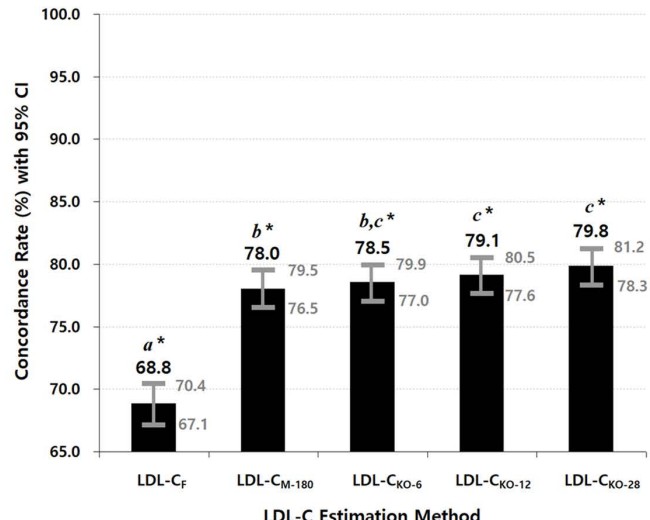

**(A) Concordance in individuals with TG < 150 mg/dL**

**(B) Concordance in individuals with TG levels of 150–399 mg/dL**

**Fig 3. Comparison of concordance rates between Friedewald and Martin–Hopkins LDL-C estimates by TG levels: <150 mg/dL and 150–399 mg/dL, based on NCEP–ATP III guideline.** Panel (A) shows results for individuals with TG levels <150 mg/dL, and Panel (B) for those with TG levels of 150–399 mg/dL. Bars represent concordance rates ± 95% confidence intervals for each LDL-C estimate. From left to right, bars correspond to the following LDL-C estimates: LDL-C$_F$, LDL-C$_{M-180}$, LDL-C$_{KO-6}$, LDL-C$_{KO-12}$, and LDL-C$_{KO-28}$. * Bars labeled with the same letters indicate no statistically significant difference in concordance between LDL-C estimates ($\alpha = 0.05$), based on pairwise comparisons using McNemar's exact test for correlated proportions (see S9 and S10 Tables). The 95% confidence intervals for concordance rates were calculated using the Clopper–Pearson exact method. **Abbreviations:** NCEP–ATP III: National Cholesterol Education Program Adult Treatment Panel III; TG: triglyceride; LDL-C: low-density lipoprotein cholesterol; LDL-C$_F$: LDL-C calculated using the Friedewald formula; LDL-C$_{M-180}$: LDL-C calculated using the original 180-cell Martin–Hopkins equation proposed by Martin et al. [14]; LDL-C$_{KO-N}$ (LDL-C$_{KO-6-TG}$, LDL-C$_{KO-12}$, and LDL-C$_{KO-28}$): LDL-C calculated using the N-cell tables with the optimal ratios of triglycerides to very-low-density lipoprotein cholesterol (TG/VLDL-C) derived from our dataset.

limits of agreement (mean ± 1.96 SD), respectively. Relative to LDL-C$_F$, both LDL-C$_{M-180}$ and LDL-C$_{KO-28}$ exhibited reduced bias and tighter limits of agreement, reflecting improved concordance with LDL-C$_D$ values. In particular, for individuals with TG levels of 300–399 mg/dL (represented by green dots), the mean bias and 95% limits of agreement (mean ± 1.96 SD) were: −13.49 ± 28.61 mg/dL for LDL-C$_F$, 6.69 ± 26.19 mg/dL for LDL-C$_{M-180}$, and 2.61 ± 25.51 mg/dL for LDL-C$_{KO-28}$. These findings provide additional evidence supporting the improved accuracy of the optimized estimation method (LDL-C$_{KO-28}$), especially among individuals with elevated TG levels.

We compared the concordance rates of three LDL-C estimates—LDL-C$_F$, LDL-C$_{M-180}$, and LDL-C$_{KO-28}$—with directly measured LDL-C across estimated LDL-C treatment categories defined by the NCEP–ATP III guideline, stratified by TG levels: <150 mg/dL and 150–399 mg/dL (Fig 5). Among individuals with TG levels <150 mg/dL, the concordance rates for LDL-C$_F$ tended to decrease as LDL-C$_F$ levels increased, declining from 84.0% for LDL-C$_F$ <70 mg/dL to 72.1% for LDL-C$_F$ ≥ 190 mg/dL. In contrast, among those with TG levels of 150–399 mg/dL, the concordance rates for LDL-C$_F$ showed an increasing trend with rising LDL-C$_F$ levels, from 47.5% for LDL-C$_F$ <70 mg/dL to 84.8% for LDL-C$_F$ ≥ 190 mg/dL (see S11 Table for details).

Compared to LDL-C$_F$, concordance improvements with LDL-C$_{M-180}$ and LDL-C$_{KO-28}$ were more pronounced when classifying lower LDL-C levels among individuals with TG levels of 150–399 mg/dL (Fig 5B). The most substantial enhancement was observed in the classification of LDL-C <70 mg/dL, where concordance increased from 47.5% with LDL-C$_F$ to 90.3% with LDL-C$_{M-180}$ and 93.1% with LDL-C$_{KO-28}$. However, in treatment categories with LDL-C ≥ 130 mg/dL within the same TG range, both LDL-C$_{M-180}$ and LDL-C$_{KO-28}$ exhibited slightly lower concordance rates than LDL-C$_F$.

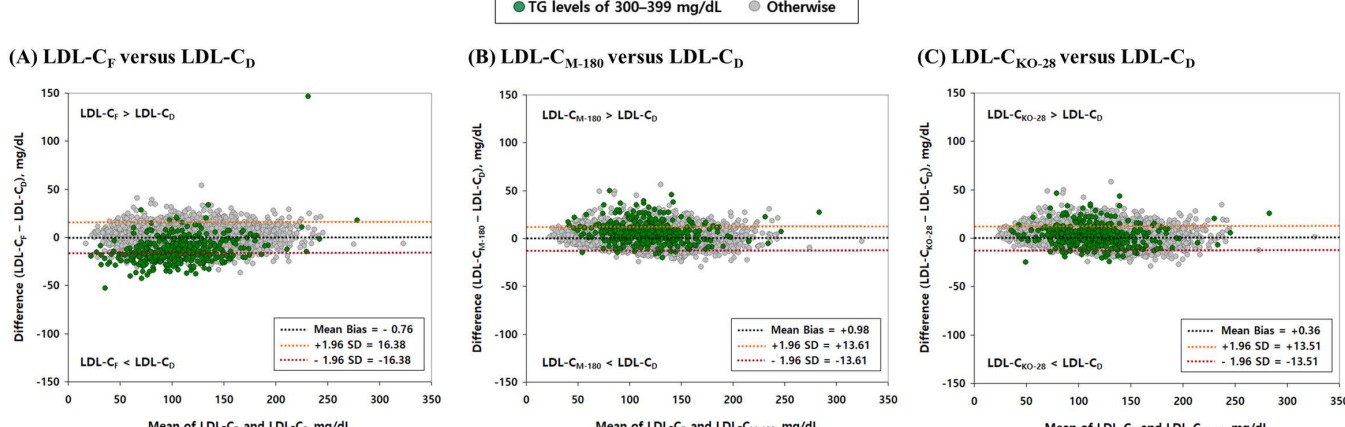

**Fig 4. Bland–Altman plots comparing estimated LDL-C values with directly measured LDL-C.** Each panel displays the bias (i.e., the difference between LDL-C$_E$ and LDL-C$_D$) plotted against the mean of LDL-C$_E$ and LDL-C$_D$. The black dashed line represents the mean bias, while the orange and red dashed lines indicate the upper and lower 95% limits of agreement (mean ± 1.96 SD), respectively. Panels (A), (B), and (C) correspond to LDL-C$_F$, LDL-C$_{M-180}$, and LDL-C$_{KO-28}$, respectively. These plots illustrate differences in bias magnitude and variability, reflecting the level of agreement between each LDL-C$_E$ and LDL-C$_D$. Green dots indicate individuals with TG levels of 300–399 mg/dL. **Abbreviations:** LDL-C: low-density lipoprotein cholesterol; TG: triglyceride; LDL-C$_E$: estimated LDL-C; LDL-C$_D$: LDL-C directly measured using a homogeneous enzymatic assay; LDL-C$_F$: LDL-C calculated using the Friedewald formula; LDL-C$_{M-180}$: LDL-C calculated using the original 180-cell Martin–Hopkins equation proposed by Martin et al. [14]; LDL-C$_{KO-28}$: LDL-C calculated using the 28-cell table (Fig 2) with the optimal ratios of triglycerides to very-low-density lipoprotein cholesterol (TG/VLDL-C) derived from our dataset; SD: standard deviation.

## Discordance by TG and estimated LDL-C levels

To evaluate classification accuracy according to the NCEP–ATP III guideline, we assessed the extent to which LDL-C treatment categories based on three estimation methods—LDL-C$_F$, LDL-C$_{M-180}$, and LDL-C$_{KO-28}$—were reclassified by LDL-C$_D$ (S12 Table). Among individuals whose treatment category changed, we identified two types of reclassification: upward (reclassified into a higher LDL-C treatment category) and downward (reclassified into a lower category) reclassification. For example, among individuals with TG levels < 400 mg/dL and estimated LDL-C levels < 70 mg/dL, upward reclassification to LDL-C ≥ 70 mg/dL occurred in 27.4% for LDL-C$_F$, 15.1% for LDL-C$_{M-180}$, and 11.8% for LDL-C$_{KO-28}$.

Fig 6 illustrates the discordance between LDL-C$_D$ and estimated LDL-C—either LDL-C$_F$ or LDL-C$_{KO-28}$—according to the NCEP–ATP III guideline classification. Discordance was categorized as under-classification (when the estimated LDL-C falls into a lower category than LDL-C$_D$) and over-classification (when it falls into a higher category), reflecting the need for upward or downward reclassification, respectively.

Fig 6A and 6B display the discordance rates of LDL-C$_F$ and LDL-C$_{KO-28}$, respectively, across TG levels. LDL-C$_F$ exhibited increasing under-classification with rising TG levels, whereas over-classification was more prevalent at lower TG levels (Fig 6A). In contrast, LDL-C$_{KO-28}$ substantially reduced under-classification at higher TG levels; however, this improvement was partially offset by increased rates of over-classification (Fig 6B). For example, in the TG range of 300–399 mg/dL, the under-classification rate for LDL-C$_{KO-28}$ was 9.0%, markedly lower than 48.4% for LDL-C$_F$, although the over-classification rate for LDL-C$_{KO-28}$ was 15.3%, notably higher than 3.2% observed with LDL-C$_F$.

Fig 6C–6F present discordance rates across LDL-C treatment categories, stratified by TG levels. Among individuals with TG levels < 150 mg/dL (Fig 6C and 6D), there was no substantial difference in the patterns of under- and over-classification between LDL-C$_F$ and LDL-C$_{KO-28}$. However, in individuals with TG levels of 150–399 mg/dL (Fig 6E and 6F), the differences between the two LDL-C estimates became more pronounced. LDL-C$_F$ showed markedly increased under-classification at lower LDL-C levels, while LDL-C$_{KO-28}$ substantially mitigated this under-classification. For instance,

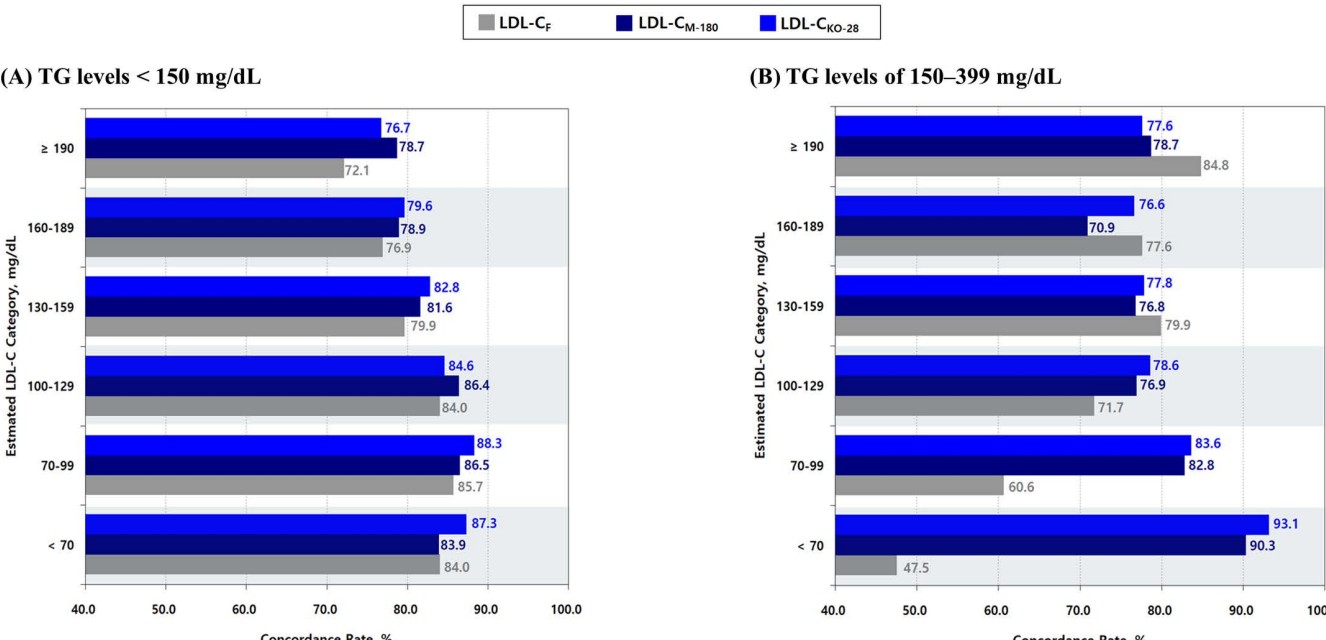

**Fig 5. Concordance in the NCEP–ATP III guideline classification: Friedewald vs. Martin–Hopkins LDL-C estimates.** The figure compares concordance rates for LDL-C estimates stratified by TG levels: < 150 mg/dL (A) and 150–399 mg/dL **(B)**. The LDL-C estimates include LDL-C$_F$ (gray), LDL-C$_{M-180}$ (dark blue), and LDL-C$_{KO-28}$ (blue). The bars indicate concordance rates for each estimated LDL-C category according to the NCEP–ATP III guideline classification, illustrating differences in concordance across the methods. **Abbreviations:** NCEP–ATP III: National Cholesterol Education Program Adult Treatment Panel III; LDL-C: low-density lipoprotein cholesterol; TG: triglyceride; LDL-C$_F$: LDL-C calculated using the Friedewald formula; LDL-C$_{M-180}$: LDL-C calculated using the original 180-cell Martin–Hopkins equation proposed by Martin et al. [14]; LDL-C$_{KO-28}$: LDL-C calculated using the 28-cell table (Fig 2) with the optimal ratios of triglycerides to very-low-density lipoprotein cholesterol (TG/VLDL-C) derived from our dataset.

the under-classification rate in classifying LDL-C levels < 70 mg/dL was 6.9% for LDL-C$_{KO-28}$, markedly lower than 52.5% observed with LDL-C$_F$, indicating a significant improvement in classification accuracy with the optimized method.

## Treatment category reclassification by LDL-C estimates

Table 5 presents the reclassification outcomes of LDL-C$_{M-180}$ and LDL-C$_{KO-28}$ across treatment categories initially assigned by LDL-C$_F$ in individuals with TG levels < 400 mg/dL, evaluated against directly measured LDL-C. Among individuals with LDL-C$_F$ < 70 mg/dL, 19.3% and 19.6% were correctly reclassified upward to LDL-C ≥ 70 mg/dL using LDL-C$_{M-180}$ and LDL-C$_{KO-28}$, respectively. However, 6.7% and 5.9% were incorrectly reclassified to LDL-C ≥ 70 mg/dL, despite being correctly classified as LDL-C < 70 mg/dL by LDL-C$_F$. For all LDL-C$_F$ categories below 160 mg/dL, both methods yielded more correct than incorrect reclassifications, leading to statistically significant improvements in classification accuracy compared to LDL-C$_F$ ($p < 0.01$ for all comparisons). Overall, LDL-C$_{KO-28}$ demonstrated slightly superior performance to LDL-C$_{M-180}$, with a higher rate of correct reclassifications (8.4% vs. 7.9%) and a lower rate of incorrect reclassifications (4.1% vs. 4.3%).

Fig 7 illustrates the proportions of correct (blue) and incorrect (gray) reclassifications using LDL-C$_{M-180}$ and LDL-C$_{KO-28}$, relative to treatment categories initially assigned by LDL-C$_F$, according to the NCEP–ATP III guideline. For both methods, reclassification was more frequent among individuals with higher TG levels (Fig 7A and 7B). In particular, for TG levels of 200–299 and 300–399 mg/dL, the proportion of correct reclassifications greatly exceeded that of incorrect ones for both methods, compared to the TG < 150 mg/dL subgroups. Notably, in the highest TG category (300–399 mg/dL), LDL-C$_{M-180}$ exhibited a slightly higher rate of correct reclassification than LDL-C$_{KO-28}$ (42.3% vs. 39.4%), but also a substantially higher rate of incorrect reclassification (20.9% vs. 12.2%).

**(A) Discordance for LDL-C$_F$ by TG levels**

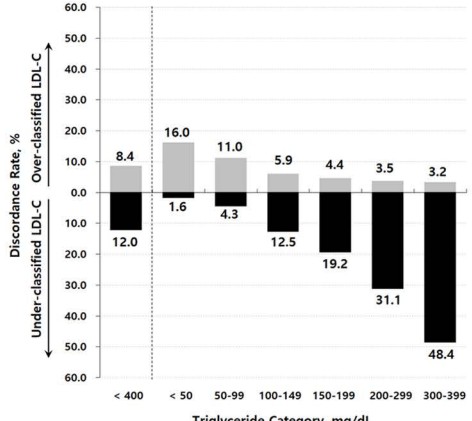

**(B) Discordance for LDL-C$_{KO-28}$ by TG levels**

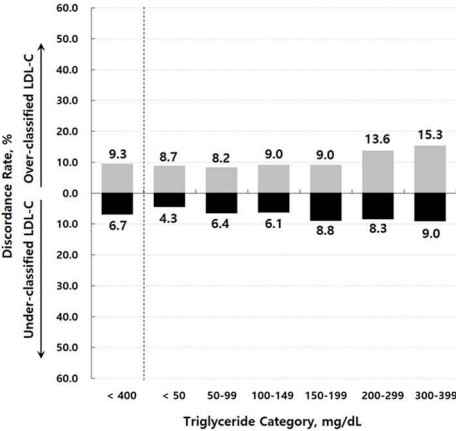

**(C) Discordance for LDL-C$_F$ by LDL-C$_F$ levels in individuals with TG levels < 150 mg/dL**

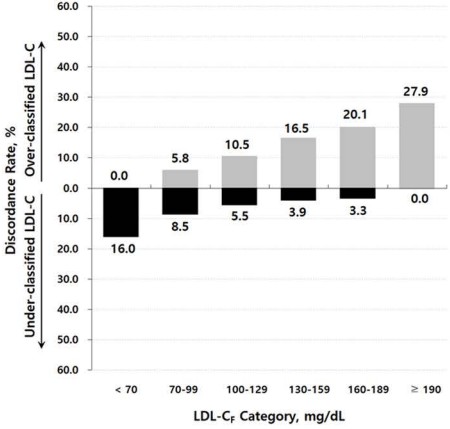

**(D) Discordance for LDL-C$_{KO-28}$ by LDL-C$_{KO-28}$ in individuals with TG levels < 150 mg/dL**

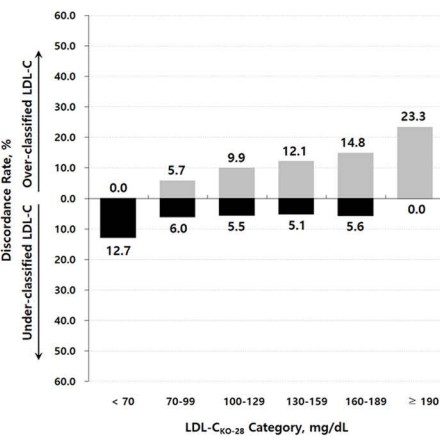

**(E) Discordance for LDL-C$_F$ by LDL-C$_F$ levels in individuals with TG levels of 150–399 mg/dL**

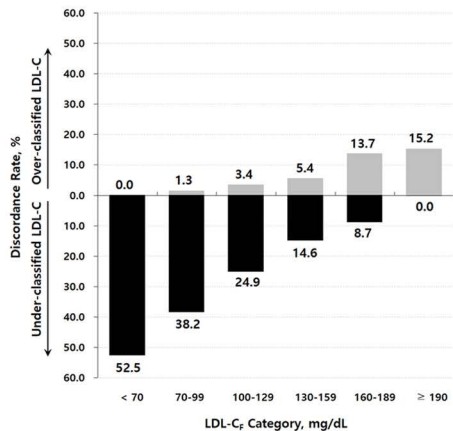

**(F) Discordance for LDL-C$_{KO-28}$ by LDL-C$_{KO-28}$ levels in individuals with TG levels of 150–399 mg/dL**

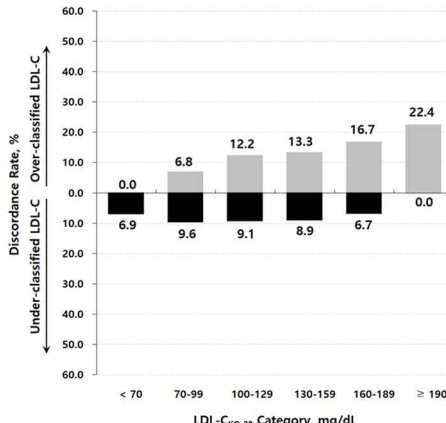

**Fig 6. Discordance rates in LDL-C classification according to the NCEP–ATP III guideline: Friedewald vs. Martin–Hopkins LDL-C estimates.**
This figure illustrates differences in discordance rates between two LDL-C estimates calculated using the Friedewald formula (LDL-C$_F$) and the optimized 28-cell Martin–Hopkins method (LDL-C$_{KO-28}$). Discordance is divided into under-classification (black bars) and over-classification (gray bars), and

is stratified by either TG levels or estimated LDL-C categories. Panels (A) and (B) show discordance rates by TG levels for LDL-C$_F$ and LDL-C$_{KO-28}$, respectively. Panels (C) and (D) display discordance by LDL-C categories in individuals with TG levels < 150 mg/dL. Panels (E) and (F) present corresponding results for individuals with TG levels of 150–399 mg/dL. This arrangement allows direct visual comparison between the two LDL-C estimates across different lipid profiles. **Abbreviations:** NCEP–ATP III: National Cholesterol Education Program Adult Treatment Panel III; TG: triglyceride; LDL-C: low-density lipoprotein cholesterol; LDL-C$_F$: LDL-C calculated using the Friedewald formula; LDL-C$_{KO-28}$: LDL-C calculated using the 28-cell table (Fig 2) with the optimal ratios of triglycerides to very-low-density lipoprotein cholesterol (TG/VLDL-C) derived from our dataset.

**Table 5. Reclassification outcomes using LDL-C$_{M-180}$ and LDL-C$_{KO-28}$ across treatment categories assigned by LDL-C$_F$, as evaluated against directly measured LDL-C in individuals with triglyceride levels < 400 mg/dL.**

| LDL-C$_F$, mg/dL | Classification | Reclassification using LDL-C$_{M-180}$ a | | | Reclassification using LDL-C$_{KO-28}$ a | | |
|---|---|---|---|---|---|---|---|
| | | Correct | Incorrect | p-value b | Correct | Incorrect | p-value b |
| < 70 | Correct | 635 (65.9) | **65 (6.7)** | < 0.001 | 643 (66.7) | **57 (5.9)** | < 0.001 |
| (n = 964) | Incorrect | **186 (19.3)** | 78 (8.1) | | **189 (19.6)** | 75 (7.8) | |
| 70–99 | Correct | 2,861 (75.4) | **188 (5.0)** | < 0.001 | 2,886 (76.1) | **163 (4.3)** | < 0.001 |
| (n = 3,793) | Incorrect | **359 (9.5)** | 385 (10.2) | | **378 (10.0)** | 366 (9.6) | |
| 100–129 | Correct | 3,186 (76.6) | **185 (4.4)** | < 0.001 | 3,235 (77.8) | **136 (3.3)** | < 0.001 |
| (n = 4,158) | Incorrect | **268 (6.4)** | 519 (12.5) | | **243 (5.8)** | 544 (13.1) | |
| 130–159 | Correct | 1,664 (76.8) | **64 (3.0)** | 0.008 | 1648 (76.0) | **80 (3.7)** | < 0.001 |
| (n = 2,168) | Incorrect | **99 (4.6)** | 341 (15.7) | | **139 (6.4)** | 301 (13.9) | |
| 160–189 | Correct | 507 (74.9) | **14 (2.1)** | 0.311 | 481 (71.0) | **40 (5.9)** | 0.456 |
| (n = 677) | Incorrect | **21 (3.1)** | 135 (19.9) | | **48 (7.1)** | 108 (16.0) | |
| ≥ 190 | Correct | 130 (76.5) | **1 (0.6)** | 0.021 | 120 (70.6) | **11 (6.5)** | 1.000 |
| (n = 170) | Incorrect | **9 (5.3)** | 30 (17.6) | | **10 (5.9)** | 29 (17.1) | |
| Overall | Correct | 8,983 (75.3) | **517 (4.3)** | < 0.001 | 9,013 (75.5) | **487 (4.1)** | < 0.001 |
| (n = 11,930) | Incorrect | **942 (7.9)** | 1,488 (12.5) | | **1,007 (8.4)** | 1,423 (11.9) | |

NCEP–ATP III: National Cholesterol Education Program Adult Treatment Panel III; LDL-C: low-density lipoprotein cholesterol; LDL-C$_F$: LDL-C calculated using the Friedewald formula; LDL-C$_{M-180}$: LDL-C calculated using the original 180-cell Martin–Hopkins equation proposed by Martin et al. [14]; LDL-C$_{KO-28}$: LDL-C calculated using the 28-cell table (Fig 2) with the optimal ratios of triglycerides to very-low-density lipoprotein cholesterol (TG/VLDL-C) derived from our dataset.

[a]Values are presented as numbers (percentages within each LDL-C$_F$ category). Boldfaced values highlight cases in which the classification by LDL-C$_{M-180}$ or LDL-C$_{KO-28}$ differed from that of LDL-C$_F$, based on directly measured LDL-C.

[b]Statistical significance of the differences in concordance between each LDL-C estimate and LDL-C$_F$ was assessed using McNemar's exact test for correlated proportions.

Fig 7C–7F present reclassification outcomes across treatment categories based on LDL-C$_F$, stratified by TG levels. Compared to individuals with TG levels < 150 mg/dL (Fig 7C and 7D), reclassification was markedly more prevalent among those with TG levels of 150–399 mg/dL (Fig 7E and 7F), particularly within lower LDL-C$_F$ categories. Among individuals with LDL-C$_F$ < 70 mg/dL and TG levels of 150–399 mg/dL, 48.8% and 49.8% were correctly reclassified by LDL-C$_{M-180}$ and LDL-C$_{KO-28}$, respectively. However, 13.6% and 11.6% were incorrectly reclassified by LDL-C$_{M-180}$ and LDL-C$_{KO-28}$, despite being correctly classified as < 70 mg/dL by LDL-C$_F$. For LDL-C$_F$ categories < 130 mg/dL, both methods yielded significantly more correct than incorrect reclassifications, whereas for LDL-C$_F$ categories ≥ 130 mg/dL, the rates of correct and incorrect reclassification were similar and no statistically significant improvement over LDL-C$_F$ was observed (see S14 Table).

Among 964 individuals classified as having LDL-C < 70 mg/dL by LDL-C$_F$, S1 Fig presents the concordance rates of LDL-C estimates calculated using the Friedewald and Martin–Hopkins methods, compared with directly measured LDL-C. This analysis follows the NCEP–ATP III guideline and is stratified by TG levels: < 150 mg/dL (S1A Fig) and 150–399 mg/dL (S1B Fig). The Martin–Hopkins-based estimates included not only LDL-C$_{M-180}$ but also LDL-C$_{KO-6}$, LDL-C$_{KO-12}$, and

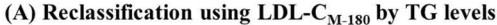

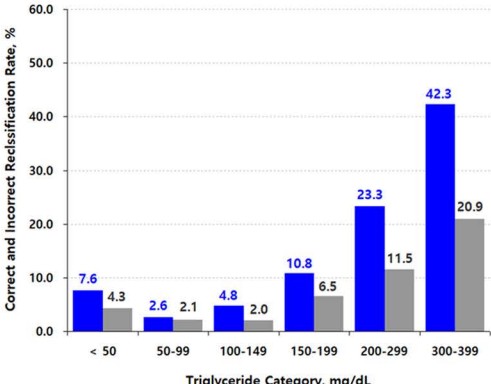

**(A) Reclassification using LDL-C$_{M-180}$ by TG levels**

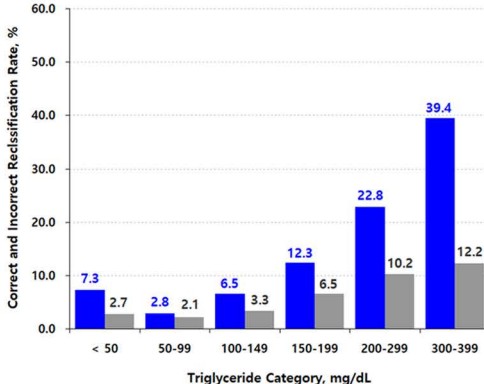

**(B) Reclassification using LDL-C$_{KO-28}$ by TG levels**

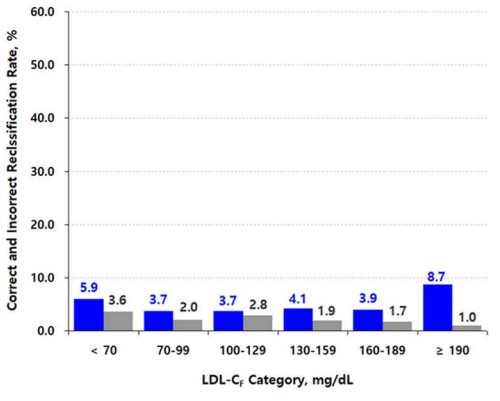

**(C) Reclassification using LDL-C$_{M-180}$ by LDL-C$_F$ levels in individuals with TG levels < 150 mg/dL**

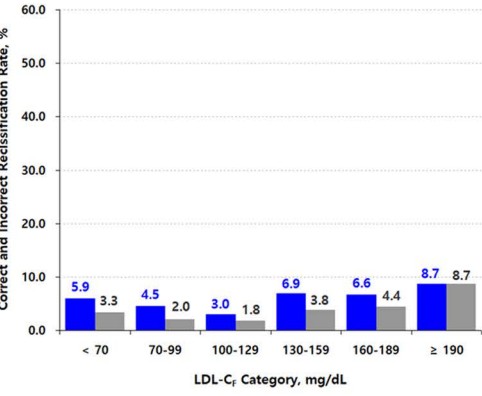

**(D) Reclassification using LDL-C$_{KO-28}$ by LDL-C$_F$ levels in individuals with TG levels < 150 mg/dL**

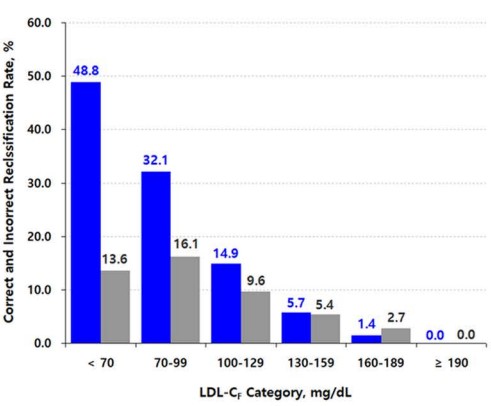

**(E) Reclassification using LDL-C$_{M-180}$ by LDL-C$_F$ levels in individuals with TG levels of 150-399 mg/dL**

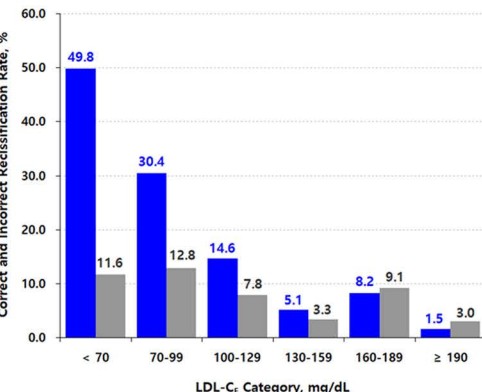

**(F) Reclassification using LDL-C$_{KO-28}$ by LDL-C$_F$ levels in individuals with TG levels of 150–399 mg/dL**

**Fig 7. Correct and incorrect reclassification using LDL-C$_{M-180}$ and LDL-C$_{KO-28}$ compared with LDL-C$_F$.** Reclassification was defined as a shift in the NCEP–ATP III guideline-based treatment category initially assigned by LDL-C$_F$ when using either LDL-C$_{M-180}$ or LDL-C$_{KO-28}$. The correctness of reclassification was evaluated against directly measured LDL-C, and each panel illustrates the proportions of correct (blue) and incorrect

(gray) reclassifications. Panels (A) and (B) show reclassification by triglyceride levels (< 50, 50–99, 100–149, 150–199, 200–299, and 300–399 mg/dL) using LDL-C$_{M-180}$ and LDL-C$_{KO-28}$, respectively. Panels (C) and (D) show reclassification by LDL-C$_F$ treatment categories among individuals with TG levels <150 mg/dL, using LDL-C$_{M-180}$ and LDL-C$_{KO-28}$, respectively. Panels (E) and (F) present the same comparisons for individuals with TG levels of 150–399 mg/dL. **Abbreviations:** NCEP–ATP III: National Cholesterol Education Program Adult Treatment Panel III; LDL-C: low-density lipoprotein cholesterol; LDL-C$_F$: LDL-C calculated using the Friedewald formula; LDL-C$_{M-180}$: LDL-C calculated using the original 180-cell Martin–Hopkins equation proposed by Martin et al. [14]; LDL-C$_{KO-28}$: LDL-C calculated using the 28-cell table (Fig 2) with the optimal ratios of triglycerides to very-low-density lipoprotein cholesterol (TG/VLDL-C) derived from our dataset.

LDL-C$_{KO-28}$, all of which were calculated using optimized TG/VLDL-C ratios derived from our dataset. The statistical significance of differences in overall concordance between these LDL-C estimates was assessed, as detailed in S15 and S16 Tables. Among individuals with TG levels < 150 mg/dL, all Martin-Hopkins-based LDL-C estimates—except LDL-C$_{M-180}$—showed statistically significant improvements in concordance compared to LDL-C$_F$; however, the magnitude of improvement was modest, with increases of less than 3%. In contrast, for individuals with TG levels of 150–399 mg/dL, all four Martin–Hopkins-based estimates demonstrated substantial improvements in concordance compared to LDL-C$_F$, with increases exceeding 30% ($p < 0.001$ for all comparisons).

## Discussion

In this study, we evaluated the performance of the Martin–Hopkins method compared to the Friedewald formula for estimating LDL-C in a sample of 18,322 Koreans with LDL-C measured by a homogeneous enzymatic assay. A subset of 11,930 subjects with TG distributions similar to the 2015 KNHSP cohort was selected to validate both methods as well as other LDL-C estimation equations. To assess the potential for improvement through the use of optimal TG/VLDL-C ratios instead of median TG/VLDL-C ratios, we calculated two variations of LDL-C estimates (LDL-C$_{KM-N}$ and LDL-C$_{KO-N}$) using strata-specific median and optimal TG/VLDL-C ratios derived from the entire sample.

Our findings can be summarized as follows: (1) The 180-cell Martin–Hopkins equation demonstrated superior concordance with directly measured LDL-C treatment categories compared to the Friedewald formula, particularly among individuals with elevated TG levels and when classifying lower LDL-C values. (2) For TG levels < 400 mg/dL, the overall concordance improved modestly but significantly (79.6% for LDL-C$_F$ vs. 83.2% for LDL-C$_{M-180}$). This modest improvement reflects that the greatest benefits were observed in individuals with high TG levels, who represent a relatively smaller proportion of the population. (3) Substituting the published median TG/VLDL-C ratios with those derived from our population led to a more accurate LDL-C estimation, emphasizing the importance of tailoring ratios to specific populations and LDL-C measurement methods. (4) Applying strata-specific optimal TG/VLDL-C ratios further enhanced estimation performance, especially when stratification was based on both TG and non–HDL-C levels. This suggests that while median ratios may suffice for TG-only stratifications, they are suboptimal for multidimensional schemes. (5) Despite using fewer stratification cells, the optimized 28-cell estimate achieved slightly higher concordance than the median-based 180-cell estimate among individuals with TG levels < 400 mg/dL (84.0% for LDL-C$_{KO-28}$ vs. 83.7% for LDL-C$_{KM-180}$), highlighting the potential benefits of using optimal TG/VLDL-C ratios even in simpler stratification structures.

### Two major approaches for LDL-C estimation

Accurate assessment of LDL-C is of broad clinical interest and remains essential for evidence-based prevention of CVDs. Although beta quantification is the reference method, its high cost and technical complexity make it impractical for routine use. The Friedewald formula—based on a fixed TG/VLDL-C ratio of 5—has long served as a convenient alternative but is known to underestimate LDL-C in individuals with elevated TG and low LDL-C concentrations, which is consistent with our findings (Fig 6E).

To address the limitations of the Friedewald formula, several alternative equations have been developed over the years. Building on Friedewald's concept of estimating LDL-C from the lipid profile, two major methodological approaches have

emerged in the literature. One approach, exemplified by the Martin–Hopkins equation [14], utilizes strata-specific TG/VLDL-C ratios that vary across lipid ranges [17,21]. The other approach, represented by the Sampson equation [15], employs multiple least squares regression to predict LDL-C using lipid parameters as independent variables [16,18,22–29].

An inherent limitation of the stratification-based approach is that increasing the number of strata requires a sufficiently large sample size to ensure stable and reliable ratio estimates, thereby restricting the number of covariates that can be incorporated when defining strata. In contrast, regression-based methods allow for the simultaneous inclusion of multiple predictors, even in relatively small samples. However, a key drawback is that regression coefficients may be highly sensitive to the characteristics of the training data—particularly the distribution of TG levels—raising concerns about their generalizability.

We focused on the strata-specific TG/VLDL-C ratio approach, as extended in the Martin–Hopkins method by incorporating both TG and non–HDL-C levels for stratification, due to its conceptual continuity with the Friedewald formula and thus its increased potential for clinical adoption. To further refine the Martin–Hopkins method, we applied optimal TG/VLDL-C ratios within each stratum to maximize concordance. However, exploring regression-based LDL-C prediction, including non-linear or machine learning models, remains a valid and promising alternative.

## Validation of the 180-cell Martin–Hopkins equation

Among the 17 LDL-C estimation equations evaluated, the original 180-cell Martin–Hopkins equation (LDL-C$_{M-180}$) demonstrated the highest overall concordance (S7 Table), although the extent of improvement varied by TG level. As illustrated in Fig 3, the improvement over the Friedewald formula (LDL-C$_F$) was modest among individuals with TG levels < 150 mg/dL (84.9% vs. 83.3%), whereas a substantial gain was observed in those with TG levels of 150–399 mg/dL (78.0% vs. 68.8%), underscoring the enhanced performance of LDL-C$_{M-180}$ in populations with higher TG levels. The most pronounced improvement was observed in classifying LDL-C levels < 70 mg/dL for individuals with TG levels of 150–399 mg/dL: 47.5% for LDL-C$_F$ vs. 90.3% for LDL-C$_{M-180}$ (Fig 5B). These results demonstrate the clinical utility of LDL-C$_{M-180}$ in accurately identifying individuals who may benefit from intensive lipid-lowering therapy.

Previous studies have reported varying differences in overall concordance between LDL-C$_F$ and LDL-C$_{M-180}$. Martin et al. [14] reported a difference of 6.3% (85.4% for LDL-C$_F$ vs. 91.7% for LDL-C$_{M-180}$), while Samuel et al. [30] observed a difference of 6.4% (83.2% vs. 89.6%). In contrast, Meeusen et al. [11] found only a minimal difference of 0.8% (76.9% vs. 77.7%). In our study, the difference in overall concordance was 3.6% (79.6% vs. 83.2%), indicating a modest yet significant improvement of LDL-C$_{M-180}$ over LDL-C$_F$.

## Impact of TG distribution on overall concordance

Variations in TG distribution are a key factor contributing to differences in overall concordance observed across studies. The median TG level in the cohort analyzed by Meeusen et al. [11] was 131 mg/dL, which is notably higher than the median levels reported in other studies: 115 mg/dL in the Martin et al. [14], 114 mg/dL in Samuel et al. [30], and 112 mg/dL in our cohort (Population 1). This elevated TG distribution in the Meeusen et al. study likely contributed to the relatively lower overall concordances observed. However, TG distribution alone does not fully account for the nearly identical overall concordance rates between LDL-C$_F$ and LDL-C$_{M-180}$ reported by Meeusen et al. [11]. Given that LDL-C$_{M-180}$ has demonstrated improved classification accuracy, particularly among individuals with elevated TG levels, a more pronounced difference in concordance would have been expected in that context.

## Differences in LDL-C measurement methods

Some of the discrepancies in overall concordance between LDL-C$_F$ and LDL-C$_{M-180}$ across the studies may stem from differences in the LDL-C measurement methods employed. Meeusen et al. [11] employed beta quantification (also referred

to as preparative ultracentrifugation, PUC), whereas Martin et al. [14] and Samuel et al. [30] used the Vertical Auto Profile (VAP) technique. In contrast, our study utilized a homogeneous enzymatic assay.

## Need for accurate assessment of low LDL-C levels

Scharnagl et al. [9] were the first to report that the Friedewald formula significantly underestimates LDL-C and may not be reliable in patients with low LDL-C levels, such as those undergoing LDL apheresis or receiving intensive lipid-lowering treatment. Consistent with this observation, our study found that among individuals with TG levels of 150–399 mg/dL classified by the Friedewald formula as having LDL-C < 70 mg/dL, 52.5% had directly measured LDL-C levels ≥ 70 mg/dL (Fig 6E). This substantial rate of underestimation suggests that a considerable proportion of high-risk individuals may be misclassified and, consequently, may not receive the level of treatment warranted by their actual lipid profile.

Given these concerns, the use of direct chemical assays is often recommended for patients with elevated TG levels or those at high cardiovascular risk [37]. However, direct measurement of LDL-C presents certain challenges, as its performance has been shown to vary depending on the manufacturer. For example, one study [34] reported that overall concordance differed across three commonly used homogeneous assays (Roche, Beckman, and Siemens), highlighting the variability and lack of standardization in direct assay methods.

As highlighted by Meeusen et al. [11], the current standard practice relies on LDL-C calculated by the Friedewald formula rather than LDL-C measured by PUC. Given the potential implications for patient care, clinicians should consider using direct LDL-C measurement methods or more accurate alternative formulas to ensure proper risk classification and therapeutic decision-making, particularly for patients with low LDL-C levels.

With the widespread adoption of detergent-based assays in routine clinical practice, our study—based on a homogeneous enzymatic assay—demonstrates a key advantage of the Martin–Hopkins method over the Friedewald formula. Specifically, the Martin–Hopkins method significantly improved the classification of low LDL-C levels among individuals with elevated TG levels (Fig 5B), highlighting its clinical relevance in high-risk populations.

Nevertheless, as Stein [38] noted, the Martin–Hopkins method should be validated using LDL-C levels determined by PUC to ensure its reliability. Meeusen et al. [11] evaluated its performance in a cohort of 23,055 patients with LDL-C measured by PUC and reported improved concordance compared to the Friedewald formula for classifying LDL-C levels < 70 mg/dL and 70–99 mg/dL. However, concordance was lower for other classification ranges, resulting in nearly identical overall concordance rates between the two equations. Consistent with this observation, our study also found that the Martin–Hopkins method did not demonstrate clear performance advantages over the Friedewald formula for treatment categories with LDL-C thresholds ≥ 130 mg/dL. In particular, among individuals with TG levels of 150–399 mg/dL, the Martin–Hopkins method showed lower concordance than the Friedewald formula in the higher LDL-C categories (S11 Table).

## Challenges with the 180-cell stratification

Although the 180-cell TG/VLDL-C ratio matrix proposed by Martin et al. [14] could be easily implemented in modern computer-based clinical practice, its exhaustive nature poses significant challenges for robust external validation. Multiple studies—including Meeusen et al. [11], our own, and others [33]—have attempted to replicate the median TG/VLDL-C ratio matrix but failed to reproduce the consistent trend reported by Martin et al. [14], wherein the ratios increased with rising TG levels and decreased with increasing non–HDL-C levels within the same TG stratum.

This inconsistency may partly explain why Meeusen et al. [11] did not proceed to evaluate the performance of LDL-C estimation, despite presenting a full 180-cell matrix derived from a large PUC-measured dataset ($n = 23,000$). Even with such a substantial sample size, stabilizing all 180 strata may have been infeasible, underscoring the practical limitations of highly granular stratification schemes.

Moreover, our findings indicate that greater granularity does not necessarily translate into improved performance. Among individuals with TG levels of 300–399 mg/dL, the concordance for LDL-C$_{M-180}$ (69.8%) was unexpectedly lower than that of the simpler 10-cell model, LDL-C$_{M-10}$ (73.0%), as shown in Table 4. This finding suggests that excessive stratification may lead to overfitting of TG/VLDL-C ratios to the derivation dataset, thereby compromising generalizability and limiting clinical applicability.

## Tailoring TG/VLDL-C ratios to specific populations

The accuracy of LDL-C estimation methods is influenced not only by population characteristics but also by the method used to measure LDL-C, underscoring the need to validate the Martin–Hopkins method using median TG/VLDL-C ratios tailored to specific study populations. In this study, we compared LDL-C estimation accuracy within the identical stratification schemes, applying the median ratios proposed by Martin et al. [14] versus those derived from our own dataset.

We first calculated two estimates—LDL-C$_{M-10}$ and LDL-C$_{M-180}$—using the median TG/VLDL-C ratios from the 10-cell and 180-cell tables, respectively, as described by Martin et al. [14]. As shown in Table 4, LDL-C$_{M-180}$ exhibited a marginal yet significant improvement in overall concordance compared to LDL-C$_{M-10}$, with a difference of 0.4% (82.8% for LDL-C$_{M-10}$ vs. 83.2% for LDL-C$_{M-180}$, $p = 0.045$). When applying the median TG/VLDL-C ratios derived from our own dataset using the same stratification schemes, the improvement in overall concordance increased to 0.9% (82.8% for LDL-C$_{KM-10}$ vs. 83.7% for LDL-C$_{KM-180}$, $p < 0.001$). Martin et al. [14] reported a comparable improvement of 1.2% (90.5% for LDL-C$_{M-10}$ vs. 91.7% for LDL-C$_{M-180}$), indicating that the use of the 180-cell Martin–Hopkins stratification scheme, compared to the 10-cell scheme, may yield an approximate 1% gain in concordance. This suggests that increasing the granularity of stratification contributes to a modest enhancement in estimation accuracy—albeit at the cost of reduced feasibility for robust external validation, as finer stratification schemes require larger sample sizes to yield stable ratio estimates.

For the 180-cell scheme, substituting the median TG/VLDL-C ratios proposed by Martin et al. [14] with those derived from our dataset resulted in more accurate LDL-C estimation (83.2% for LDL-C$_{M-180}$ vs. 83.7% for LDL-C$_{KM-180}$, $p = 0.003$), as shown in Table 4. These findings highlight the critical role of tailoring TG/VLDL-C ratios to specific populations and LDL-C measurement methods to enhance estimation accuracy. In contrast, for the 10-cell scheme, there was no difference in overall concordance between LDL-C$_{M-10}$ and LDL-C$_{KM-10}$ (both 82.8%). This suggests that adopting a coarser but appropriate level of stratification may enhance the generalizability of the estimation method, even if it entails a slight compromise in estimation accuracy.

## Stratification schemes and parameter selection

Building on the observation that stratification granularity affects LDL-C estimation accuracy, we further examined how specific design choices—such as the selection of stratification parameters, the determination of cutoff points, and the number of strata—impact the performance of the Martin–Hopkins method.

To assess the effect of stratification based solely on TG levels, we constructed two models—LDL-C$_{KM-6-TG}$ and LDL-C$_{KM-12-TG}$—using the 6-cell (first column of Table 3) and 12-cell (S2 Table) stratification schemes, respectively. As shown in Table 4 and S9 Table, increasing the number of TG-based strata did not result in a significant improvement in overall concordance (83.0% for LDL-C$_{KM-6-TG}$ vs. 83.1% for LDL-C$_{KM-12-TG}$, $p = 0.226$).

To further investigate the impact of adding non–HDL-C as a second stratification parameter, we developed two additional models—LDL-C$_{KM-12}$ and LDL-C$_{KM-28}$—based on the 12-cell (middle and last columns of Table 3) and 28-cell (Fig 2) schemes, respectively. These models added non–HDL-C cutoffs in addition to the six TG-based strata. Compared to the 6-cell model based solely on TG levels (83.0% for LDL-C$_{KM-6-TG}$), the 12-cell model yielded no significant improvement (83.2% for LDL-C$_{KM-12}$, $p = 0.162$). In contrast, the 28-cell model demonstrated a statistically significant enhancement in concordance (83.4% for LDL-C$_{KM-28}$, $p = 0.016$).

These findings suggest that merely increasing the number of TG-based strata provides limited benefit and may yield diminishing returns. However, appropriately incorporating additional lipid parameters—such as non–HDL-C—into the stratification scheme can improve estimation accuracy when applied with sufficient granularity. This reflects the heuristic nature of the Martin–Hopkins method and underscores the importance of empirical exploration and population-specific optimization in the selection of stratification parameters.

## Median vs. optimal TG/VLDL-C ratios

The original Martin–Hopkins method employs strata-specific median TG/VLDL-C ratios for estimating VLDL-C. However, it remains unclear whether these median statistics provide the best representation of TG/VLDL-C ratios for accurate LDL-C estimation. To investigate this, we derived strata-specific optimal TG/VLDL-C ratios—ranging from 1.0 to 10.0 in 0.1-unit increments—to maximize concordance between estimated and directly measured LDL-C levels according to the NCEP–ATP III guideline classification.

As anticipated, the use of optimal ratios improved overall concordance relative to median-based estimates. However, in TG–only stratifications, these improvements were not statistically significant: 83.0% for LDL-$C_{KM-6-TG}$ vs. 83.2% for LDL-$C_{KO-6-TG}$ ($p = 0.089$) and 83.1% for LDL-$C_{KM-12-TG}$ vs. 83.4% for LDL-$C_{KO-12-TG}$ ($p = 0.087$). These findings suggest that median ratios remain a viable option for TG-only stratifications.

In contrast, when both TG and non–HDL-C levels were used for stratification, the LDL-$C_{KO-N}$ estimates consistently outperformed their median-based counterparts, the LDL-$C_{KM-N}$ estimates (Table 4). This supports the utility of optimizing TG/VLDL-C ratios in multidimensional stratification schemes and highlights a practical refinement to the Martin–Hopkins method that may enhance accuracy without excessive stratification.

## LDL-C risk reclassification and implications

Our study demonstrates that refining the stratification-based Martin–Hopkins method by applying optimal TG/VLDL-C ratios can improve concordance with directly measured LDL-C categories. To address the clinical relevance of this improvement, we conducted a reclassification analysis to examine how many individuals would move across clinical decision thresholds when using either LDL-$C_{M-180}$ or LDL-$C_{KO-28}$ instead of the Friedewald formula (LDL-$C_F$).

Reclassification was particularly notable in individuals with LDL-$C_F$ < 70 mg/dL, where 19.3% (LDL-$C_{M-180}$) and 19.6% (LDL-$C_{KO-28}$) were correctly reclassified to LDL-C ≥ 70 mg/dL (Table 5), suggesting potential underestimation by LDL-$C_F$ that could affect pharmacologic treatment eligibility. This trend was more pronounced in individuals with TG levels of 150–399 mg/dL, a group prone to misclassification, where nearly half were correctly reclassified (48.8% and 49.8%, respectively), and only 13.6% and 11.6% were incorrectly reclassified upward (Fig 7E and 7F). Among individuals with LDL-$C_F$ categories < 130 mg/dL and TG levels of 150–399 mg/dL, both methods resulted in significantly more correct than incorrect reclassifications, leading to statistically significant improvements in classification accuracy compared to LDL-$C_F$ (S14 Table).

These findings underscore the importance of employing more accurate LDL-C estimation methods, particularly for patients with low LDL-$C_F$ and elevated TG levels. Implementing LDL-$C_{M-180}$ and LDL-$C_{KO-28}$ may enable clinicians to more appropriately identify patients who require lipid-lowering therapy or closer clinical follow-up. This improvement in classification accuracy is not only statistically significant but also clinically meaningful.

## Distinct modeling approach and generalizability

Although the original Martin–Hopkins method was developed using a large U.S. cohort ($n > 900,000$), our study—based on a Korean cohort ($n = 18,322$)—demonstrated that population-specific optimization can yield superior performance. Importantly, our approach does not merely recalibrate the original Martin–Hopkins equation. Rather, it represents a distinct

model that optimizes TG/VLDL-C ratios to enhance concordance with directly measured LDL-C values. For example, despite employing fewer stratification cells, the 28-cell optimized LDL-C$_{KO-28}$ achieved slightly higher concordance (84.0%) than the 180-cell median-based LDL-C$_{KM-180}$ (83.7%) among individuals with TG levels < 400 mg/dL (Table 4). This finding suggests that careful optimization can outperform more granular stratification schemes based on median values, thereby enabling the use of simpler, less granular stratifications. Such models may offer greater flexibility for incorporating novel variables that could further enhance performance. As larger and more diverse datasets become available, the optimized approach can be extended to include additional parameters (e.g., HDL-C, obesity-related metrics) beyond TG and non–HDL-C, further improving the accuracy of LDL-C estimation.

### Assay-specific and ethnic considerations

LDL-C measurement techniques may influence the TG/VLDL-C ratios derived for estimation. Our study employed a homogeneous enzymatic assay, while the original Martin–Hopkins method was developed using the VAP technique, which separates lipoprotein fractions through a different process. These methodological differences can significantly affect TG/VLDL-C ratio estimation and, consequently, the performance of LDL-C estimation methods. This highlights the need for assay-specific recalibration when applying such models in diverse laboratory settings. In addition, ethnic and genetic factors, along with lifestyle and dietary patterns, may influence lipid profiles, including TG/VLDL-C ratios, across populations. While our optimized model improved concordance in a Korean cohort, external validation in diverse ethnic groups and clinical settings is essential to support broader applicability and generalizability across populations.

### Practical implementation and scalability

Many laboratories routinely collect lipid profile data, making local optimization of TG/VLDL-C ratios both feasible and potentially impactful. The appropriate level of stratification should be guided by the size of the available dataset. Our findings demonstrate that even a simplified 6-cell model based solely on TG levels (LDL-C$_{KO-6-TG}$) can substantially improve concordance over LDL-C$_F$, particularly among individuals with TG levels of 150–399 mg/dL (Fig 3B). When sufficient data are available, incorporating stratification by non–HDL-C to form a 12-cell model (LDL-C$_{KO-12}$) can enhance performance. However, increasing the number of strata should be approached with caution. As the number of cells or strata increases, the statistical stability and reliability of the derived TG/VLDL-C ratios may decline, and the resulting performance gains may be only marginal. In settings where local recalibration is not feasible, validated ratio tables from demographically or methodologically comparable populations may serve as a practical alternative. Nonetheless, as assay platforms and lipid distributions evolve over time, periodic reassessment and updating of TG/VLDL-C ratios should be integrated into routine quality control practices, particularly when transitioning to new laboratory assays.

### Study limitations

This study has several limitations. First and foremost, we did not use beta quantification, the reference method for LDL-C measurement. Due to its large-scale and logistical constraints, beta quantification was not feasible in the KNHANES. Instead, the KNHANES employed the Sekisui homogeneous LDL-C assay, which has been widely adopted in national health surveys and routine clinical laboratories across Korea. Although a previous study [39] reported that the Sekisui assay failed to meet the NCEP–ATP III total allowable error (TAE) criterion of ±12% in certain diseased populations, more recent evidence from a multicenter study by Miida et al. [40] demonstrated that the assay satisfied the NCEP–ATP III criteria, showing a total error of 10.9%, an imprecision (total coefficient of variation, CV$_t$) of 4.7%, and a mean bias of 1.0% among diseased individuals with TG levels < 400 mg/dL—consistent with our study population. Therefore, while we acknowledge the absence of the reference method, we believe that the analytical performance of the Sekisui homogeneous assay is sufficient to justify its use in this context. Nonetheless, our findings should be interpreted with this limitation

in mind. Second, the TG/VLDL-C ratios derived from our cohort may not be generalizable to populations with differing demographic or metabolic characteristics. While our findings highlight the importance of tailoring TG/VLDL-C ratios to specific populations, the ratios may not be universally applicable. Third, the sample size of our study was insufficient to fully implement the 180-cell stratification using optimal TG/VLDL-C ratios. This limitation restricted our ability to evaluate the potential benefits of more granular stratification for improving LDL-C estimation accuracy. Finally, the cross-sectional nature of our study precludes long-term assessment of LDL-C estimation consistency. Changes in TG levels and metabolic conditions over time could influence TG/VLDL-C ratios, thereby affecting the performance of LDL-C estimation methods.

## Conclusion

This study addresses a critical issue in cardiovascular risk assessment by evaluating the accuracy of LDL-C estimation methods, with a particular focus on the Martin–Hopkins method and its potential for improvement through the use of optimal TG/VLDL-C ratios. While the Martin–Hopkins method—based on strata-specific median TG/VLDL-C ratios—has demonstrated superior accuracy in previous studies, its reliance on median statistics raises questions regarding its optimality. Using data from 18,322 individuals in the KNHANES, we derived strata-specific optimal TG/VLDL-C ratios aimed at maximizing concordance with directly measured LDL-C values according to the NCEP–ATP III guideline classification. Our findings reveal that applying optimal TG/VLDL-C ratios within the Martin–Hopkins method enhances LDL-C estimation accuracy, particularly when stratification incorporates both TG and non–HDL-C levels. Importantly, this improvement can be achieved without increasing the number of strata, offering a practical pathway to refine LDL-C estimation while avoiding excessive stratification. Efforts to refine and expand the Martin–Hopkins method may benefit from leveraging optimal TG/VLDL-C ratios to explore parameter spaces more effectively. This approach enables precise tailoring of TG/VLDL-C ratios to factors such as lipid profiles, insulin resistance, obesity, and age—each of which may influence VLDL-C variability.

## Supporting information

**S1 Fig. Concordance rates of LDL-C estimates using Friedewald and Martin–Hopkins methods among individuals with LDL-C$_F$ < 70 mg/dL, compared with directly measured LDL-C, according to the NCEP–ATP III guideline.** Bars represent concordance rates ± 95% confidence intervals for LDL-C estimates, stratified by triglyceride levels of < 150 mg/dL (Panel A) and 150–399 mg/dL (Panel B). From left to right, bars correspond to the following LDL-C estimates: LDL-C$_F$, LDL-C$_{M-180}$, LDL-C$_{KO-6}$, LDL-C$_{KO-12}$, and LDL-C$_{KO-28}$. * Bars labeled with the same letters indicate no statistically significant difference in concordance between LDL-C estimates ($\alpha$ = 0.05), based on pairwise comparisons using McNemar's exact test for correlated proportions (see S15 and S16 Tables for details). The 95% confidence intervals for concordance rates were calculated using the Clopper–Pearson exact method. **Abbreviations:** NCEP–ATP III: National Cholesterol Education Program Adult Treatment Panel III; TG: triglyceride; LDL-C: low-density lipoprotein cholesterol; LDL-C$_F$: LDL-C calculated using the Friedewald formula; LDL-C$_{M-180}$: LDL-C calculated using the original 180-cell Martin–Hopkins equation proposed by Martin et al. [14]; LDL-C$_{KO-N}$ (LDL-C$_{KO-6-TG}$, LDL-C$_{KO-12}$, and LDL-C$_{KO-28}$): LDL-C calculated using the N-cell tables for the optimal TG/VLDL-C ratios derived from our dataset.
(TIF)

**S1 Table. Comparison of triglyceride distributions between the study population and participants in the 2015 Korea National Health Screening Program (KNHSP).**
(DOCX)

**S2 Table. Median (95% CI) and optimal TG/VLDL-C ratios by triglyceride strata (12-cell).**
(DOCX)

**S3 Table. Median (95% CI) and optimal TG/VLDL-C ratios by combined triglyceride and non–HDL-C strata (10-cell).**
(DOCX)

**S4 Table. Median TG/VLDL-C ratios by combined triglyceride and non–HDL-C strata (180-cell).**
(DOCX)

**S5 Table. Median (95% CI) and optimal TG/VLDL-C ratios by combined triglyceride and non–HDL-C strata (28-cell).**
(DOCX)

**S6 Table. Statistical significance of differences in overall concordance between LDL-C estimates among individuals with triglyceride levels < 400 mg/dL.**
(DOCX)

**S7 Table. Concordance in the NCEP–ATP III guideline classification: Friedewald, Martin–Hopkins, and other LDL-C estimates compared to directly measured LDL-C, overall and by triglyceride strata.**
(DOCX)

**S8 Table. Model fit and error metrics for estimated LDL-C (LDL-C$_E$) versus directly measured LDL-C (LDL-C$_D$): Comparison of Friedewald, Martin–Hopkins, and other estimation methods.**
(DOCX)

**S9 Table. Statistical significance of differences in overall concordance between LDL-C estimates among individuals with triglyceride levels < 150 mg/dL.**
(DOCX)

**S10 Table. Statistical significance of differences in overall concordance between LDL-C estimates among individuals with triglyceride levels of 150–399 mg/dL.**
(DOCX)

**S11 Table. Concordance in the NCEP–ATP III guideline classification: Friedewald versus Martin–Hopkins estimates of low-density lipoprotein cholesterol (LDL-C) compared to direct LDL-C, overall and by TG strata.**
(DOCX)

**S12 Table. Concordance and discordance in the NCEP–ATP III guideline classification: Friedewald vs. Martin–Hopkins estimates of low-density lipoprotein cholesterol (LDL-C) compared to directly measured LDL-C.**
(DOCX)

**S13 Table. Reclassification outcomes using LDL-C$_{M-180}$ and LDL-C$_{KO-28}$ across treatment categories assigned by LDL-C$_F$, evaluated against directly measured LDL-C in individuals with TG levels < 150 mg/dL.**
(DOCX)

**S14 Table. Reclassification outcomes using LDL-C$_{M-180}$ and LDL-C$_{KO-28}$ across treatment categories assigned by LDL-C$_F$, evaluated against directly measured LDL-C in individuals with TG levels of 150–399 mg/dL.**
(DOCX)

**S15 Table. Statistical significance of differences in overall concordance between LDL-C estimates among individuals with triglyceride levels < 150 mg/dL and LDL-C$_F$ < 70 mg/dL.**
(DOCX)

**S16 Table. Statistical significance of differences in overall concordance between LDL-C estimates among individuals with triglyceride levels of 150–399 mg/dL and LDL-C$_F$ <70 mg/dL.**
(DOCX)

## Author contributions

**Conceptualization:** Jongseok Lee, Jun Seok, In Cheol Jeong.

**Data curation:** Jongseok Lee, Hyelim Lee, Hwajung Cha.

**Formal analysis:** Jongseok Lee, Hyelim Lee, Hwajung Cha.

**Funding acquisition:** Jongseok Lee, In Cheol Jeong.

**Investigation:** Jongseok Lee, Hyelim Lee, Hwajung Cha, Jun Seok.

**Methodology:** Jongseok Lee, Hyelim Lee, Hwajung Cha, Jun Seok.

**Project administration:** Jongseok Lee, In Cheol Jeong.

**Resources:** Jun Seok, In Cheol Jeong.

**Software:** Jongseok Lee, Hyelim Lee, Hwajung Cha.

**Supervision:** Jongseok Lee.

**Validation:** Jongseok Lee, Jun Seok, In Cheol Jeong.

**Visualization:** Jongseok Lee, Hyelim Lee, Hwajung Cha.

**Writing – original draft:** Jongseok Lee, Hyelim Lee, Hwajung Cha, Jun Seok.

**Writing – review & editing:** Jongseok Lee, In Cheol Jeong.

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
