## [Decision Letter · Decision Letter 0]

14 Apr 2025

Thank you for submitting your manuscript to PLOS ONE. After careful consideration, we feel that it has merit but does not fully meet PLOS ONE’s publication criteria as it currently stands. Therefore, we invite you to submit a revised version of the manuscript that addresses the points raised during the review process.

We look forward to receiving your revised manuscript.

Kind regards,

Misbahuddin Rafeeq

Academic Editor

PLOS ONE

Journal Requirements:

“This work was supported by a National Research Foundation of Korea (NRF) grant funded by the Korean Government (MSIT) (No. 2022R1A5A8019303).”

Reviewers' comments:

Reviewer's Responses to Questions

**Comments to the Author**

1. Is the manuscript technically sound, and do the data support the conclusions?

Reviewer #1: Partly

Reviewer #2: Yes

Reviewer #3: Yes

Reviewer #4: Partly

2. Has the statistical analysis been performed appropriately and rigorously?

Reviewer #1: Yes

Reviewer #2: Yes

Reviewer #3: Yes

Reviewer #4: Yes

3. Have the authors made all data underlying the findings in their manuscript fully available?

Reviewer #1: Yes

Reviewer #2: Yes

Reviewer #3: Yes

Reviewer #4: Yes

4. Is the manuscript presented in an intelligible fashion and written in standard English?

Reviewer #1: Yes

Reviewer #2: Yes

Reviewer #3: Yes

Reviewer #4: Yes

Reviewer #1: The authors tried to answer an important question of quantifying low-density lipoprotein cholesterol (LDL-C) using the calculation method, especially for the decision points. The manuscript is well written. However, they should address the following queries.

1. It is a record-based study; thus, using beta quantification for LDL-C estimation is implausible. However, a homogenous method (Sekisui Medical) was employed. In a previous study (10.1373/clinchem.2009.142810), this method failed to achieve the total allowable error (TAE) as per the NCEP guidelines in LDL-C quantification among diseased subjects, though it worked well in normal people. Based on this fact, kindly answer the following points:

A) How do you justify this method in the absence of the gold standard, as your data must have included diseased subjects?

B) Kindly mention the total error with imprecision and bias, separately related to your data.

2. Include Bland-Altman plots for the concordance study.

Reviewer #2: The authors compare two methods of calculating low-density lipoprotein cholesterol (LDL-C), which is a key risk factor for cardiovascular disease, bringing into question recent findings that the Martin-Hopkins method is superior to the Friedewald method. The manuscript is well written, clear, and assertive, and the authors clearly state the sources of their data as well as their methods of analysis.

Some specific points:

-In the abstract, it is unclear what the connection is between the two methods and whether they are both for low TG levels. I recommend parsing that out more explicitly, as well as the strengths of the Martin-Hopkins method. For exampe, when it is stated that the method is successful in estimating very low-density lipoprotein cholesterol, it is unclear what that is compared to 400 mg/dL. Later in the manuscript, it seems that very low-density lipoprotein cholesterol is referring to lower levels, less than 70-79 mg/dL. I think it would be helpful to delineate that distinction more clearly early on.

-When showing the formulas for Friedewald and Martin-Hopkins, it is states "where, AF_N is an adustable factor...", remove the comma, so it is "where AF_N is an adustable factor."

-In the results, it is states "Table 1 summarized" - perhaps it should be "Table 1 summarizes"? Keeping it consistent that the analyses performed are described in the past, but the tables are currently showing results.

Reviewer #3: Summary:

This study evaluates the Martin-Hopkins method for estimating LDL-C in a Korean population and shows that tailoring TG/VLDL-C ratios to the local population improves accuracy over both the traditional Friedewald formula and published median-based methods. The greatest benefit is seen in individuals with high triglycerides and low LDL-C levels. The study also finds that excessive stratification offers limited gains and may reduce generalizability.

Major Comments:

- Can the authors discuss the applicability of the findings for global population 9since this only use korean population)?

- Can the authors add a discussion section on how the improved concordance would affect patient classification or treatment decisions—e.g., how many patients would move across clinical thresholds?

- While tailoring TG/VLDL-C ratios to specific populations improves estimation accuracy, how feasible is it to implement this approach in clinical practice? Could the authors discuss practical implementation strategies—such as whether hospitals and laboratories can use population-specific median tables or if periodic recalibration would be necessary to maintain accuracy across different settings?

Minor:

- Some parts of the text repeat the same points in different sections—for example, the limitations of the Friedewald formula in people with high triglycerides come up multiple times. It would help to streamline the writing by removing repetition and combining related results into a single summary table or figure for better clarity.

Reviewer #4: Summary

The Martin-Hopkins equation for LDL estimation has been shown to be more effective at calculating LDL compared to the Friedwald equation, which has been the standard of care. The MH equation uses a median TG/VLDL-C ratio in different strata of TG and Non HDL Cholesterol. The authors propose two changes to the MH equation, learning strata-specific median TG/VLDL-C ratios in a specific cohort of patients in Korea, and also optimizing this value for maximum concordance within stratas.

Major Comments

Methods

- Rather than optimizing for concordance within each stratum, why not directly fit the linear equation to predict LDL-C? Is it due to a lack of sample size? Mentioning why or why not that is possible would be beneficial.

Results

- There should be error bars, confidence intervals, p-values, or some measure of statistical significance when comparing concordance

- It may be informative to see evaluation done for more than just concordance. The review by Samuel et al (2023) of 23 equations for LDL-C that showed the MH equation to be the best had a 3-stage testing scheme that included concordance, MAE, stratification across demographic groups, and, lastly, clinical classification

Ref: Samuel C, Park J, Sajja A, Michos ED, Blumenthal RS, Jones SR, Martin SS. Accuracy of 23 Equations for Estimating LDL Cholesterol in a Clinical Laboratory Database of 5,051,467 Patients. Glob Heart. 2023 Jun 19;18(1):36. doi: 10.5334/gh.1214. PMID: 37361322; PMCID: PMC10289049

Discussion

- One aspect to highlight may be the difference in sample size for the training of the MH equation and the specific equation in the Korean sample. If you can outperform the MH-equation with a smaller subset of training data, there is potential for even better prediction with a larger sample size to train on. Specifically, when talking about using the optimal TG/VLDL-C ratio rather than median, this coul be considered a different formula completely, rather than just refitting the MH Equation within your own sample

Minor Comments

‘this improvement can achieve without increasing the number of strata,’ in the conclusion has a typo, should be ‘this improvement can be achieved without increasing the number of strata,’

A key on what the colors mean in figure 2 would be helpful

**Do you want your identity to be public for this peer review?** For information about this choice, including consent withdrawal, please see our Privacy Policy

Reviewer #1: No

Reviewer #2: No

Reviewer #3: No

Reviewer #4: No

---

## [Author Response · Author response to Decision Letter 1]

29 May 2025

Response to Reviewer 1 Comments

We sincerely appreciate your encouraging and insightful comments, which helped us to better clarify and refine the key points of our manuscript. Below, we repeat your points and address them individually.

Point 1: It is a record-based study; thus, using beta quantification for LDL-C estimation is implausible. However, a homogenous method (Sekisui Medical) was employed. In a previous study (10.1373/clinchem.2009.142810), this method failed to achieve the total allowable error (TAE) as per the NCEP guidelines in LDL-C quantification among diseased subjects, though it worked well in normal people. Based on this fact, kindly answer the following points:

(A) How do you justify this method in the absence of the gold standard, as your data must have included diseased subjects?

(B) Kindly mention the total error with imprecision and bias, separately related to your data.

Response 1 (A):

We appreciate the reviewer’s insightful comments regarding the absence of the beta-quantification reference method in our study and the implications of using the Sekisui homogeneous LDL-C assay. We fully acknowledge that beta quantification remains the gold standard reference method for measuring LDL-C. However, due to logistical constraints and the large-scale nature of the Korea National Health and Nutrition Examination Survey (KNHANES), the use of beta quantification was not feasible. Instead, KNHANES employed a homogeneous enzymatic assay (Sekisui Medical), which has been widely adopted in routine clinical laboratories and national health surveys in Korea.

As the reviewer correctly noted, Miller et al. (2010) reported that the Sekisui homogeneous method did not meet the National Cholesterol Education Program (NCEP) total allowable error (TAE) criterion of ±12% in certain diseased populations. Nevertheless, among the homogeneous assays evaluated, it showed one of the lowest total errors (13.5%) and the smallest negative mean bias (–1.7%).

Importantly, a more recent multicenter study by Miida et al. (2012) confirmed that the Sekisui assay demonstrated good agreement with the beta-quantification method and satisfied the NCEP total error criterion (< 12%) in both non-diseased and diseased populations. Specifically, within the diseased group, the reported total errors by triglyceride (TG) level were:

• 11.5% for TG levels < 11.29 mmol/L,

• 11.1% for TG levels < 6.78 mmol/L, and

• 10.8% for TG levels < 4.52 mmol/L.

These findings support the appropriateness of using the Sekisui assay in large-scale, population-based studies like ours.

Response 1 (B):

Thank you for this important request. To address the reviewer’s request regarding the total error and its components, we refer to the results of the multicenter evaluation study by Miida et al. (2012), which assessed the analytical performance of the Sekisui homogeneous enzymatic method across various TG levels and clinical conditions.

Because our analysis was restricted to participants with TG levels < 400 mg/dL (4.52 mmol/L), we cite the corresponding subgroup results for diseased individuals. According to Table 2 of Miida et al. (2012), the analytical performance metrics under these conditions were as follows:

• Imprecision (CVt): 4.7%

• Mean bias: 1.0%

• Total error (TE): 10.9%

The total error value of 10.9% falls within the NCEP recommended threshold of < 12%, even in the diseased subgroup. Therefore, considering that our study population included only individuals with TG levels < 400 mg/dL, we believe that the use of the Sekisui assay is analytically justified and reliable for assessing LDL-C estimation methods in this context.

To address this issue, we have added the following clarification to the Study limitations section on pages 44–45:

This study has several limitations. First and foremost, we did not use beta quantification, the reference method for LDL-C measurement. Due to the large-scale nature and logistic constraints of the KNHANES, beta quantification was not feasible. Instead, KNHANES employed the Sekisui homogeneous LDL-C assay, which has been widely adopted in national health surveys and routine clinical laboratories across Korea. Although a previous study [39] reported that the Sekisui assay did not meet the NCEP total allowable error (TAE) criterion of ±12% in some diseased populations, more recent evidence from a multicenter study by Miida et al. [40] demonstrated that the assay satisfied the NCEP criteria, with a total error of 10.9%, an imprecision (total coefficient of variation, CVt) of 4.7%, and a mean bias of 1.0% in diseased individuals with TG levels < 400 mg/dL—consistent with our study population. Therefore, while we acknowledge the absence of the reference method, we believe that the analytical performance of the Sekisui homogeneous assay justifies its use in this context. Nonetheless, our findings should be interpreted with this limitation in mind.

We hope that this clarification adequately addresses the reviewer’s concern and reinforces the validity of our methodological approach within the practical constraints of a large-scale epidemiological study.

Point 2: Include Bland–Altman plots for the concordance study.

Response 2:

We thank you for this valuable suggestion. In response, we have included Bland–Altman analyses to assess the agreement between estimated LDL-C (LDL-CE) and directly measured LDL-C (LDL-CD). This analysis was conducted for three estimation methods: the Friedewald formula (LDL-CF), the original 180-cell Martin–Hopkins equation (LDL-CM-180), and our optimized 28-cell model (LDL-CKO-28). For each method, the bias (i.e., the difference between LDL-CE and LDL-CD) was plotted against the mean of the two values, and the mean bias as well as the 95% limits of agreement (mean ± 1.96 SD) were calculated.

These analyses have been incorporated into the Materials and Methods (Data Analysis) and Results sections, and are illustrated in the newly added Figure 4. We specifically highlighted individuals with triglyceride (TG) levels of 300–399 mg/dL (indicated by green dots) because among the Martin–Hopkins-based LDL-C estimates, LDL-CM-180 showed the lowest concordance in this TG range. For this subgroup, the mean bias and 95% limits of agreement were: −13.49 ± 28.61 mg/dL for LDL-CF, 6.69 ± 26.19 mg/dL for LDL-CM-180, and 2.61 ± 25.51 mg/dL for LDL-CKO-28. These results demonstrate improved agreement with directly measured LDL-C when using our optimized method.

We believe that the inclusion of Bland–Altman plots provides a valuable visual representation of agreement and reinforces the validity of our findings. Figure 4 has been appended to this response letter for the reviewer’s convenience.

Changes Made:

• Materials and Methods – Data Analysis (page 11):

Bland–Altman analysis was performed to assess the agreement between LDL-CE and LDL-CD. For three estimation methods (LDL-CF, LDL-CM-180, and LDL-CKO-28), the bias (difference between LDL-CE and LDL-CD) was plotted against the mean of the two values. The mean bias and the 95% limits of agreement (mean ± 1.96 SD) were calculated to evaluate systematic differences and dispersion across the range of LDL-C values. In addition, model fit and error metrics were assessed for each LDL-C estimation method in comparison with LDL-CD, including the coefficient of determination (R2), mean relative error (MRE), mean absolute error (MAE), and mean squared error (MSE).

• Results – Figure 4 (pages 23–24):

To further assess the agreement between estimated LDL-C (LDL-CE) and directly measured LDL-C (LDL-CD), Bland–Altman analyses were conducted for three estimation methods: LDL-CF, LDL-CM-180, and LDL-CKO-28. As shown in Fig 4, each panel displays the bias (i.e., the difference between LDL-CE and LDL-CD) plotted against the mean of the two values. The black dashed line indicates the mean bias, while the orange and red dashed lines represent the upper and lower 95% limits of agreement (mean ± 1.96 SD), respectively. Compared to LDL-CF, both LDL-CM-180 and LDL-CKO-28 exhibited reduced bias dispersion and narrower limits of agreement, indicating improved concordance with LDL-CD values. In particular, for individuals with TG levels of 300–399 mg/dL (represented by green dots), the mean bias and 95% limits of agreement (mean ± 1.96 SD) were: −13.49 ± 28.61 mg/dL for LDL-CF, 6.69 ± 26.19 mg/dL for LDL-CM-180, and 2.61 ± 25.51 mg/dL for LDL-CKO-28. These findings provide additional evidence supporting the improved accuracy of the optimized estimation method (LDL-CKO-28), especially among individuals with elevated TG levels.

• Results – Figure 4 legend (page 24):

Fig 4. Bland–Altman plots comparing estimated LDL-C values with directly measured LDL-C. Each panel shows the bias (i.e., the difference between LDL-CE and LDL-CD) plotted against the mean of LDL-CD and LDL-CE. The black dashed line represents the mean bias, while the orange and red dashed lines indicate the upper and lower 95% limit of agreement (mean ± 1.96 SD), respectively. Panels (A), (B), and (C) correspond to LDL-CF, LDL-CM-180, and LDL-CKO-28, respectively. These plots illustrate differences in bias magnitude and dispersion, reflecting the degree of agreement between each LDL-CE and LDL-CD. Green dots represent individuals with TG levels of 300–399 mg/dL. Abbreviations: LDL-C: low-density lipoprotein cholesterol; TG: triglyceride; LDL-CE: estimated LDL-C; LDL-CD: LDL-C directly measured using a homogeneous enzymatic assay; LDL-CF: LDL-C calculated using the Friedewald formula; LDL-CM-180: LDL-C calculated using the original 180-cell Martin–Hopkins equation proposed by Martin et al. [14]; LDL-CKO-28: LDL-C calculated using the 28-cell table (Fig 2) with the optimal ratios of TG/VLDL-C (triglycerides to very-low-density lipoprotein cholesterol) derived from our dataset; SD: standard deviation.

We hope this addition meets the reviewer’s request and strengthens the manuscript by offering a robust graphical assessment of estimation accuracy.

Response to Reviewer 2 Comments

Thank you very much for your encouragement and thoughtful comments, which stimulated us to rethink the key issues more clearly. Below, we repeat your points and address them individually.

Point 1: In the abstract, it is unclear what the connection is between the two methods and whether they are both for low TG levels. I recommend parsing that out more explicitly, as well as the strengths of the Martin–Hopkins method. For example, when it is stated that the method is successful in estimating very-low-density lipoprotein cholesterol, it is unclear what that is compared to 400 mg/dL. Later in the manuscript, it seems that very-low-density lipoprotein cholesterol is referring to lower levels, less than 70-79 mg/dL. I think it would be helpful to delineate that distinction more clearly early on.

Response 1:

We appreciate the reviewer’s thoughtful comments. In response, we have revised the manuscript to (1) clarify the performance of LDL-C estimation methods by triglyceride (TG) levels, (2) explicitly define VLDL-C to avoid potential confusion, and (3) enhance the concordance analysis across LDL-C treatment thresholds.

1. Clarifying performance by TG levels

In the previous manuscript, TG levels were stratified into six categories (Table 4), and reported that the original 180-cell Martin–Hopkins equation (LDL-CM-180) consistently outperformed the Friedewald formula (LDL-CF), particularly at higher TG levels.

To better clarify differences in concordance at lower TG levels, we have now added Figure 3 to the Results section. This figure compares the concordance of LDL-C estimates, stratified into two TG categories: < 150 mg/dL and 150–399 mg/dL. The analysis included not only LDL-CF and LDL-CM-180, but also three optimized Martin–Hopkins estimates—LDL-CKO-6, LDL-CKO-12, and LDL-CKO-28—calculated using optimal TG/VLDL-C ratios derived from our dataset. For TG levels < 150 mg/dL, the Martin–Hopkins estimates yielded statistically significantly higher concordance than LDL-CF, but the improvement was modest, at around 2%. In contrast, for TG levels of 150–399 mg/dL, the improvements were substantial, with concordance differences reaching approximately 10%.

2. Clarification of VLDL-C definition

Regarding the interpretation of VLDL-C, we acknowledge the ambiguity and have revised the manuscript to clarify that VLDL-C in our study does not refer to low LDL-C levels. Instead, it represents a residual cholesterol component calculated by subtracting HDL-C and LDL-C from total cholesterol (i.e., VLDL-C = non–HDL-C − LDL-C). This definition is consistent with standard practice in lipid profiling practice, where VLDL-C represents cholesterol carried by remnant lipoproteins, not a subclass of LDL-C. This clarification is now clearly presented in the Materials and Methods section.

3. Enhanced concordance by LDL-C treatment thresholds

We also refined our stratified concordance analysis based on estimated LDL-C thresholds as defined by the NCEP-ATP III guideline (< 70, 70–99, 100–129, 130–159, 160–189, ≥ 190 mg/dL). In Fig 5, we compared three LDL-C estimates—LDL-CF, LDL-CM-180, and LDL-CKO-28—against directly measured LDL-C within treatment categories, stratified by TG levels: < 150 mg/dL and 150–399 mg/dL. These analyses provide a more explicit understanding of when and where the Martin–Hopkins method offers the greatest benefit.

We hope that these revisions adequately address the reviewer’s concerns and enhance the clarity and clinical relevance of our findings. The revised figures have been incorporated into the manuscript and are also appended to this response letter for your convenience.

Changes Made:

• Abstract – Results (pages 2–3): We have added a sentence clarifying concordance differences by TG level to better reflect the stratified performance of the estimation methods. Given the word limit, the revision was restricted to a single sentence.

Results

The Martin–Hopkins method showed significantly higher concordance than the Friedewald formula for TG levels < 400 mg/dL (79.6% for LDL-CF vs. 83.2% for LDL-CM-180, p < 0.001). Concordance improved by less than 2% for TG levels < 150 mg/dL (83.3% vs. 84.9%), but by approximately 10% for TG levels of 150–399 mg/dL (68.8% vs. 78.0%). The largest discrepancy was observed in classifying LDL-C levels < 70 mg/dL among individuals with TG levels of 150–399 mg/dL (47.5% for LDL-CF vs. 90.3% for LDL-CM-180). However, the overall concordance differed only modestly between the 10-cell and 180-cell Martin–Hopkins equations (82.8% for LDL-CM-10 vs. 83.2% for LDL-CM-180, a difference of 0.4%), indicating only a marginal benefit despite the substantial increase in the number of strata. Using optimal TG/VLDL-C ratios increased overall concordance compared to median ratios within the same stratification, with LDL-CKO-N estimates outperforming their LDL-CKM-N counterparts. However, this improvement was not statistically significant in LDL-C estimates derived from TG-only stratification.

• Materials and Methods – Lipid measurements and LDL-C estimation (page 8): To address the confusion regarding the interpretation of VLDL-C, we revised the manuscript to clearly define its calculation.

Blood samples were collected from each participant’s antecubital vein after an overnight fast of at least 8 hours. Serum lipid concentrations were directly measured using a homogeneous enzymatic assay with an automated analyzer (Hitachi Automatic Analyzer 7600, Hitachi, Tokyo, Japan). The measurements included TC (Pureauto S CHO-N; Sekisui Medical, Tokyo, Japan), HDL-C (Cholestest N HDL; Sekisui Medical), LDL-C (Cholestest N LDL; Sekisui Medical) and triglycerides (Pureauto S TG-N; Sekisui Medical). non–HDL-C was calculated as TC minus HDL-C, and VLDL-C as non–HDL-C minus LDL-CD.

• Results – Figure 3 (pages 21–23): We have added Figure 3 to compare the concordance of LDL-C estimates, stratified by TG levels (< 150 mg/dL and 150–399 mg/dL), with detailed descriptions of the observed differences provided.

Fig 3 illustrates the concordance rates of LDL-C estimates calculated using the Friedewald and Martin–Hopkins methods, compared with directly measured LDL-C, based on the NCEP-ATP III guideline classification. The results are str

---

## [Editor Report · Decision Letter 1]

10 June 2025

Refining the Martin-Hopkins method for estimating low-density lipoprotein cholesterol levels: Median versus optimal TG/VLDL-C ratio

PONE-D-24-57824R1

Dear Dr. Jongseok Lee,

We’re pleased to inform you that your manuscript has been judged scientifically suitable for publication and will be formally accepted for publication once it meets all outstanding technical requirements.

Kind regards,

Misbahuddin Rafeeq

Academic Editor

PLOS ONE

---

## [Editor Report · Acceptance letter]

PONE-D-24-57824R1

PLOS ONE

Dear Dr. Lee,

I'm pleased to inform you that your manuscript has been deemed suitable for publication in PLOS ONE. Congratulations! Your manuscript is now being handed over to our production team.

Kind regards,

on behalf of

Dr. Misbahuddin Rafeeq

Academic Editor

PLOS ONE